



# Subsurface flow paths in a chronosequence of calcareous soils: impact of soil age and rainfall intensities on preferential flow occurrence

Anne Hartmann[1], Markus Weiler[2], Konrad Greinwald[2], and Theresa Blume[1]

[1]Section Hydrology, GFZ German Research Centre for Geosciences, Potsdam, Germany
[2]Faculty of Environment and Natural Resources, Chair of Hydrology, University of Freiburg, Freiburg, Germany

**Correspondence:** Anne Hartmann (aha@gfz-potsdam.de)

**Abstract.** Soil hydrologic processes play an important role in the hydro-pedo-geomorphological feedback cycle of landscape evolution. Soil properties and subsurface flow paths both change over time, but due to lack of observations subsurface water flow paths are often not properly represented in soil and landscape evolution models. We investigated the evolution of subsurface flow paths across a soil chronosequence in the calcareous glacier forefield at the Griessfirn glacier in the Swiss

Alps. Young soils developed from calcareous parent material usually have a high pH-value, which likely affects vegetation development and pedogenesis and thus the evolution of subsurface flow paths. We chose four glacial moraines of different ages (110, 160, 4900, and 13500 years) and conducted sprinkling experiments with the dye tracer Brilliant Blue on three plots at each moraine. Each plot was divided into three equal subplots and dyed water was applied with three different irrigation intensities (20, 40, and 60 mm h$^{-1}$) and an irrigation amount of 40 mm. Subsequent excavation of soil profiles enabled the

tracing of subsurface flow paths. A change in flow types with increasing moraine age was observed from a rather homogeneous matrix flow at 110 and 160 years to heterogeneous matrix and finger flow at 4900 and 13500 years. However, the proportion of preferential flow paths is not necessarily directly related to the moraine age, but rather to soil properties such as texture, soil layering, organic matter content and vegetation characteristics such as root length density and biomass. Irrigation intensity had an effect on the number of finger flow paths at the two old moraines. We also found that flow paths in this calcareous material

evolved differently compared to a previous study in siliceous material, which emphasizes the importance of parent material for flow path evolution. Our study provides a rare systematic data set and observations on the evolution of vertical subsurface flow paths in calcareous soils, which is useful to improve their representation in the context of landscape evolution modeling.



## 1 Introduction

Pedogenesis and its interaction with hydrological, geomorphological and ecological processes plays an important and complex role in landscape evolution. Understanding the complex interaction between soil and water during landscape evolution is crucial for the understanding of natural history, but also for the application of modern landscape management practices such as

landscape restoration. The parent material as the starting point for soil development has a strong impact on the resulting soil characteristics (Retzer, 1949). The mineral composition of the parent material determines the mineral composition of the soil and nutrients released during weathering, which can either be processed by flora and fauna or leached from the soil. The parent material influences weathering rates as well as chemical and physical properties of the soil and thus soil formation (Sauer et al., 2015) and also vegetation composition (Michalet et al., 2002).

The role of water in pedogenesis is particularly multi-facetted (Lin, 2003) and often governed by feedback cycles: Hydrology has a strong influence on the evolution of soil structure and texture, which in turn has an influence on soil hydrology. The hydraulic conditions influence geochemical and biological (weathering-) processes (White and Blum, 1995). Soil water also serves as a transport medium and therefore influences the relocation of substances and particles within the soil matrix. Finally, soil water availability, which is also governed by the soil hydraulic conditions, influences vegetation composition and coverage.

Preferential flow (which is known to be ubiquitous) further complicates the investigation of these interactions as it introduces additional spatial variability and leads to locally increased water availability, flow velocities and deeper water transport - all of which will increase rates of weathering and transport in these locations. Preferential flow is, however, not only controlled by soil properties (Koestel and Jorda, 2014) but also by rainfall characteristics such as amount (Bachmair et al., 2009) and intensity (Wiekenkamp et al., 2016). Extreme rainfall events can exert a particularly strong influence on water partitioning, and

thus affect pedogenic (e.g. weathering, solute transport) and geomorphic (e.g. water erosion) processes (van der Meij et al., 2018).

Despite the importance of the hydrological processes in this context, their consideration within Soil-Landscape Evolution Models (SLEMs) is still insufficient (Samouëlian et al., 2012). Landscape evolution modeling is an important tool to better understand landscape formation but also to assess how landscapes are affected by global climate change (Montagne and

Cornu, 2010) or anthropogenic interventions (Cui et al., 2021) such as land use change and hydrologic alterations (Newman et al., 2017). Especially, the evolution of vertical and horizontal subsurface flow paths including the occurrence of preferential flow paths are currently omitted in SLEMs (van der Meij et al., 2018), which can lead to incorrectly modeled transport and relocation processes (Sauer et al., 2012). One of the reasons for this neglect is the lack of knowledge of how soil hydraulic systems evolve over time. For a proper consideration of hydrologic processes in SLEMs more field data on the evolution of

soil hydraulic parameters and hydraulic function is needed (van der Meij et al., 2018).

Whereas many studies have investigated soil development (e.g. Egli et al. (2010), He and Tang (2008), Dümig et al. (2011), Vilmundardóttir et al. (2014), D'Amico et al. (2014) ), only a limited number of studies have investigated the evolution of soil hydraulic properties and subsurface flow paths. In previous field studies on the development of soil hydraulic properties, the main focus was on the saturated conductivity ($K_{sat}$), which was found to decrease with age and changes were mainly related



to changes in soil properties such as texture and bulk density (Brooks and Richards (1993), Beerten et al. (2012), Maier et al. (2020)). Beerten et al. (2012) found that hydraulic parameters describing the retention curves were harder to interpret and were affected by the organic carbon content. Lohse and Dietrich (2005) estimated points on the retention curves and hydraulic conductivity functions for a soil chronosequence from 300 to 4.1 million years of age developed from volcanic deposits. A shift

to higher water retention and horizontal subsurface flow was observed alongside the development from a rather homogeneous to a layered system with increasing clay content. A similar change in hydrological pathways was found in volcanic catchments (ranging in age from 0.225 to 82.2 Ma), where pathways changed from deep percolation to shallow subsurface flow (Yoshida and Troch, 2016). These studies examined the development of flow paths on a rather large spatial and temporal scale and thus did not investigate vertical preferential flow paths. A detailed investigation of vertical subsurface hydrological flow paths in

co-evolution with soil (hydraulic) properties was so far only carried out for a chronosequence of 10000 years on siliceous glacial till (Hartmann et al., 2020a), where an increase in preferential flow with increasing moraine age was found.

Previous studies showed conflicting results regarding the influence of rainfall intensity on preferential flow path occurrence. An increase in preferential flow with increasing rainfall intensity at the catchment scale was observed by evaluating the sequence of water content responses at different depths (Lin and Zhou (2008), Wiekenkamp et al. (2016), Demand et al. (2019)).

Bromid tracer experiments in lysimeters with stony soils by Cichota et al. (2016) showed an increase in preferential flow with increasing irrigation intensity under soil moisture conditions near field capacity (comparing rainfall events of 5 mm h$^{-1}$ and 20 mm h$^{-1}$). Wu et al. (2015) found, based on Brilliant Blue dye-tracer experiments, a decrease in preferential flow at higher intensities (comparing rainfall events of 50 mm h$^{-1}$, 100 mm h$^{-1}$, and 150 mm h$^{-1}$) when the initial soil water content was high. However, as the relationship between flow paths and rainfall intensity can be influenced by various factors, cross-study

comparisons are difficult and probably unlikely to identify the underlying controls. It is therefore crucial to investigate this relationship systematically and under controlled conditions.

Our study, carried out in the Swiss Alps, focuses on two main research gaps: (i) the evolution of vertical subsurface flow paths during soil formation in calcareous soils and (ii) the impact of rainfall intensities on preferential flow occurrence as soils and hillslopes evolve over time. Calcareous soils are here of special interest as they make up a third of the earth's land surface area

(Taalab et al., 2019). The high pH-value of young calcareous soils determines plant nutrient availability (low P availability, high N loss (Hopkins and Ellsworth (2005), Taalab et al. (2019))) and thus vegetation development (Michalet et al., 2002). Also weathering rates and pedogenesis in calcareous soils differ strongly from other parent materials (Musso et al. (2022), Ehrlich et al. (1955)). To investigate the temporal evolution of subsurface flow paths over the millennia, irrigation experiments were conducted on a set of four pro-glacial moraines with each moraine representing a different age class (chronosequence

approach). We used Brilliant Blue dye to trace the occurrence of vertical subsurface flow paths across 13500 years of landscape evolution recorded in our chronosequence and investigated a possible connection between the proportion of preferential flow paths to properties of the vegetation cover (e.g. coverage, root length density, biomass). With our study we generated a rare systematic data set and observations on the evolution of preferential flow paths in the context of landscape evolution, which will help to ensure proper handling of (subsurface) hydrologic processes and their role within the feedback cycle of the

hydro-pedo-geomorphological system when it comes to soil and landscape evolution modeling.



## 2 Material and methods

### 2.1 Study site

The study area at the Griessfirn glacier forefield is located above the treeline between 2030-2200 m a.s.l. in the Central Swiss Alps (46° 85'N, 8° 82'E). The geology is dominated by schists, marl, and quartzites (Musso et al., 2019), but also includes

limestone (Frey, 1965). The closest official weather station at a similar elevation (2106 m a.s.l.) is located at a distance of 48 km at Mount Pilatus (46° 98'N, 8° 25'E). The recorded annual mean temperature is 1.8 °C and the annual precipitation is 1752 mm (1981-2010) (MeteoSwiss, 2020).

A chronosequence of four moraines was selected to investigate the effect of hillslope age on subsurface flow paths. The four selected moraines were dated by Musso et al. (2019) based on historical maps and additional radiocarbon dating. The youngest

moraine is located at 2200 m a.s.l. and was dated to an age of 110 years (110a, a=years). The three other moraines are 160 (160a), 4900 (4.9ka, k=1000) , and 13500 (13.5ka) years old and are located at an elevation of 2030 m a.s.l. (see Figure 1). At each moraine the vegetation was mapped (Greinwald et al., 2021b) and the soil physical characteristics were identified in 10, 30, and 50 cm depth (Hartmann et al., 2020b).

The vegetation coverage of the two young moraines is sparse with carpet forming dwarf shrubs (e.g. *Dryas octopetala, Salix*

*retusa*) stabilizing the slopes by their dense root stocks and facilitating the establishment of other plant species such as patches of pioneer plants (e.g. *Saxifraga aizoides*) and mosses (e.g *Tortella densa, Distichium capillaceum*) at the 110a moraine and *Anthyllis vulneraria, Saxifraga oppositifolia*, or *Silene acaulis* (pioneer plants) at the 160a moraine. The soil at the 110a and 160a moraine was classified as a Hyperskeletic Leptosol (Musso et al., 2019). At the 110a moraine the soil texture was identified as mainly sandy loam with tendencies to sandy clay loam, whereas the soil type at the 160a old moraine was mainly

identified as sandy loam and in some occasions as loamy sand. The bulk density and porosity are homogeneous in the upper 50 cm at the 110a moraine and range between 1.6-1.9 g cm$^{-3}$ and around $\approx 0.3$, respectively. The 160a moraine has a slightly lower bulk density ($\approx 1.55$ g cm$^{-3}$) and higher porosity ($\approx 0.4$), but both moraines have a similarly low organic matter content (<10 weight-%). The two older moraines (4.9ka and 13.5ka) are densely covered with grass (e.g. *Festuca violacea agg.*), dwarf shrubs (e.g. *Rhododendrum hirsutum, Vaccinium myrtillus, Vaccinium vitis-idaea*) and sedges (*Carex ferruginea,*

*Carex sempervirens*). At both moraines the soil was classified as a Calcaric Skeletic Cambisol (Musso et al., 2019). The soil texture varies over the top 50 cm. Silty clay and silty clay loam is dominant in the upper 10 cm, while loam and silty loam is predominant in 30 cm, and sandy loam in 50 cm. The soil at the 13.5ka moraine contains in the top 50 cm a little less clay, but a little more silt than the soil at the 4.9ka. The organic matter content in the top 10 cm at both old moraines is with $\approx 25$ weight-% distinctly higher compared to the young moraines and declines with depth. Correspondingly, the porosity is with

$\approx 0.8$ distinctly higher and also rapidly declines with depth at both moraines, whereas the bulk density is low ($\approx 0.5$ g cm$^{-3}$) in the top soil and increases with depth, but still remains < 1.5 g cm$^{-3}$ in 50 cm.





**Table 1.** Main plot characteristics and vegetation parameters at the tracer experiment plots (BM= above ground biomass, RLD= root length density, n.a.= analysis was not carried out).

| Moraine age [years] | Plot # | Vegetation Cover [%] | Slope [°] | Species Richness | RLD [km m$^{-3}$] | BM [kg m$^{-2}$] |
|---|---|---|---|---|---|---|
| 110 | 1 | 25 | 40 | 10 | 348.6 | n.a. |
| 110 | 2 | 15 | 21 | 10 | 315.1 | n.a. |
| 110 | 3 | 55 | 25 | 13 | 150.6 | n.a. |
| 160 | 1 | 50 | 42 | 11 | 293.2 | 4.7 |
| 160 | 2 | 20 | 35 | 11 | 201.4 | 5.7 |
| 160 | 3 | 75 | 23 | 13 | 165.9 | 3.9 |
| 4 900 | 1 | 90 | 23 | 21 | 939.8 | 5.5 |
| 4 900 | 2 | 90 | 27 | 21 | 822.9 | 9.6 |
| 4 900 | 3 | 100 | 25 | 33 | 1235.16 | 8.7 |
| 13 500 | 1 | 85 | 27 | 25 | 654.2 | 5.0 |
| 13 500 | 2 | 100 | 36 | 25 | 1197.2 | 3.6 |
| 13 500 | 3 | 80 | 37 | 27 | 608.7 | 4.6 |

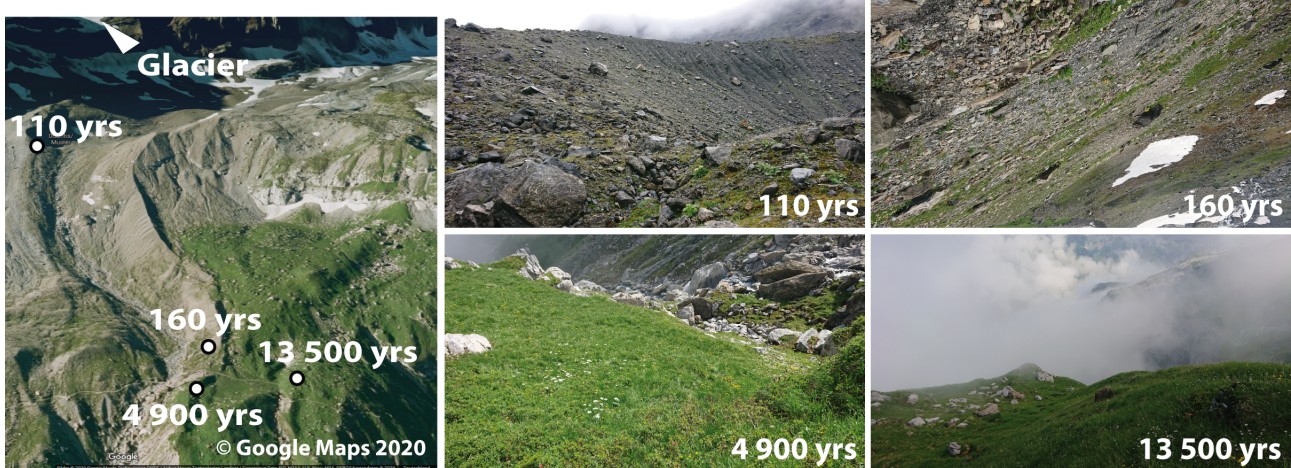

**Figure 1.** Location of the four selected moraines in the Griessfirn glacier forefield (left, photo by © Google Maps (2020)) and the surface cover of each age class (right).

For the Brilliant Blue tracer experiments, three plots (1.0 x 1.5 m) were selected on each moraine. To capture the spatial heterogeneity of the vegetation cover, the plots at each moraine were chosen along a gradient in vegetation complexity (Greinwald et al., 2021b). At each plot species richness, root length density (RLD), vegetation cover, above ground biomass (BM), and the slope were measured or estimated by Greinwald et al. (2021a) (Table 1).





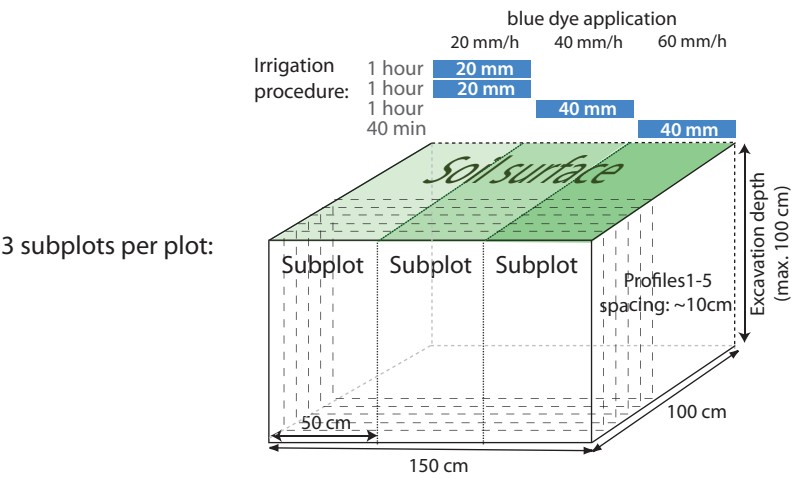

**Figure 2.** Illustration of the dye tracer experimental design at each moraine.

## 2.2 Dye tracer irrigation experiments

The dye tracer experiments were conducted between July 25th and September 14th, 2019. Each of the three 1.5 m x 1.0 m experimental plots per moraine was further divided into three equal subplots of 0.5 m x 1.0 m for individual irrigation with 40 mm of a 4 g $l^{-1}$ Brilliant Blue FCF solution (Figure 2). To reduce interception, large vegetation in form of shrubs, bushes,

and tall grass was cut to a height of a few centimeters before irrigation. The three subplots were irrigated with the same amount (40 mm) but different intensities (20, 40, and 60 mm $h^{-1}$). The irrigation intensities represent extreme events with return periods of 2.8, 60, and 100 years (Fukutome et al., 2017), respectively. Each subplot was irrigated individually, while the other two were covered by a tarpaulin. A hand-operated sprayer and a battery-powered pump were used for tracer application. Since the irrigation system only provided a flow rate of 1 l $min^{-1}$ the different irrigation intensities were achieved by alternating

intervals of irrigation and breaks. The first subplot was irrigated for 2 hours in a sequence of 1 minute irrigation and 5 minutes break to irrigate the subplot with 40 mm at an intensity of 20 mm $h^{-1}$. The intensity of 40 mm $h^{-1}$ at the second subplot was achieved by a sequence of 1 minute irrigation and 2 minutes break for 60 minutes. The last plot was irrigated for 40 minutes in a sequence of 1 minute irrigation and 1 minute break to achieve an intensity of 60 mm $h^{-1}$. An overview of the experimental design and an illustration of the irrigation procedure is provided in Figure 2. After the experiment the whole plot was covered

with a tarp to protect it from potential natural rainfall until the excavation on the following day.

A first profile was excavated 10-15 cm below the lower edge of the irrigated plot to check for subsurface lateral flow. The plots were then excavated in five vertical profiles in approximately 10 cm segments starting the first segment 10 cm from the lower edge of the irrigated plot. Pickaxes, spades, and hand shovels were used to excavate the profiles. The profiles were cleaned carefully and protruding roots were cut off. Rocks and stones were not removed, but cleaned from soil. The soil profiles of





the subplots were photographed with a Panasonic Lumix DMC-FZ18 camera and a resolution of 2248 pixels x 3264 pixels. To avoid direct sunlight and to provide a uniform light distribution a large umbrella was used for shading. A Kodak-gray-scale and a wooden frame were included in the photographs for a later geometric correction and color adjustment.

### 2.3 Image analysis and flow type classification

The photographs of the profiles were converted into tricolor images to differentiate between stained and unstained areas, as well as rocks by using the image analysis procedure by Weiler and Flühler (2004). The analysis includes a geometric correction, a background subtraction, and a color adjustment to correct differences in image illumination and changes in the spectral composition of the daylight. A further correction of the tricolor images using the photographs was necessary, since due to poor lighting conditions or a heterogeneous background color distribution in the soil caused by e.g. material transitions, small

stones or organic matter, the image analysis software was not able to recognize all large dye stains as coherent objects. Thus, the software detected interruptions within blue stains that did not correspond to the field observations and would have been identified as a large number of individual flow paths during the following analysis. The manual correction of the tricolor images using the original photographs eliminated these interruptions (Hartmann et al., 2020a). In the resulting tricolor images the horizontal and vertical length of a pixel correspond to 1 mm.

The volume density (VD) corresponds to the dye coverage and was calculated for each of the five profiles per subplot as the fraction of stained pixels in each pixel row, thus providing depth profiles of volume density. The surface area density (SAD) is an indicator for the amount of individual flow paths and was calculated for each pixel row of the five profiles by using the intercept density, which describes the number of interfaces between stained and unstained pixels divided by the horizontal width of the soil profile. The combination of both profile parameters provides the information whether the stained area is the

sum of many small fragments or a few large ones.

The dye patterns of each profile per subplot were then classified into flow type categories according to the approach proposed by Weiler and Flühler (2004). This classification is based on the proportions of three selected stained path width (SPW) classes (stained path width <20 mm, 20 mm-200 mm, >200 mm) on the volume density. The stained path width is equal to the horizontal extent of a stained flow path. This classification method distinguishes between five flow types: (1) macropore

flow with low interaction, (2) mixed macropore flow (low and high interaction), (3) macropore flow with high interaction, (4) heterogeneous matrix flow/finger flow, and (5) homogeneous matrix flow. Dye patterns, which could not be classified as one of these flow types were categorized as undefined. We used a modified version (Hartmann et al., 2020a) of this classification which was more suitable for stony alpine soils. The modified classification avoids that homogeneously blue stained areas are classified as smaller stained path widths due to the interruption by rocks. In this case the flow type is assigned to a new flow type

class called 'homogeneous matrix flow between rocks'.The modified classification also avoids a clear differentiation between 'macropore flow with high interaction' and 'finger flow'. This is based on the observation that finger-like flow paths with smaller widths were frequently present in these alpine soils which otherwise would have been misclassified as 'macropore flow with high interaction'. The classification was done for each pixel row per profile. To quantify the proportion of preferential flow per profile, a preferential flow fraction index (PFF) was calculated as the proportion of all preferential flow type classes





(1-4) at each profile.

To determine a representative infiltration depth per subplot, we used the median value of the maximum staining depths per subplot. To obtain the distribution of maximum staining depths we determined for each pixel column of the five profile images per subplot the location of the deepest blue colored pixel. The median value was chosen to represent the infiltration depth, since this measure is less affected by outliers (e.g. single deep infiltration flow paths) than the mean value.

## 2.4 Statistical analysis

### 2.4.1 Depth-dependent probability of staining

One way to describe the dye coverage per subplot is to average all five individual volume density (VD) profiles. However, the averaging over all profiles has the disadvantage that the result could be strongly influenced by large individual deviations. Another way to describe the dye coverage while accounting for the variability in the VD-profiles at each plot is to calculate a depth-dependent probability that a pixel will be stained ($p_{dye}(z)$) according to Kramers et al. (2009). In this approach the $p_{dye}$ was calculated based on a logistic regression model that calculates the log odds of a pixel in a row to be dyed according to $\ln(\frac{p_{dye}}{1-p_{dye}}) = \alpha + \beta \cdot z$, with z=depth. Each observed VD-profile per plot was transferred into a log odds profile (with odds=number of stained pixels/number of unstained pixels) and a linear mixed model was fitted (Bates et al., 2015) by including the five profiles replicates as random effects to estimate the coefficients $\alpha$ and $\beta$. The probability $p_{dye}(z)$ was then calculated by $p_{dye}(z) = \frac{1}{1+exp(-1 \cdot (\alpha+\beta \cdot z))}$.

### 2.4.2 Statistical differences between profiles

A bootstrapped LOESS regression (BLR) approach (Keith et al., 2016) was used to test differences between experiments in the observed VD-profiles and SAD-profiles with regard to moraine age and irrigation intensity. The BLR approach is a combination of bootstrapped data resampling with local least-squares-based polynomial smoothing (LOESS) regression and was proposed by Keith et al. (2016) for the comparison of any soil property profiles from data sets of two different characteristics. For the pair-wise test the two data sets containing several profile observations of the soil variable of interest are combined and depth-wise resampled n=1000 times by bootstrapping with replacement. Each resampled data set is modelled using LOESS regression. Out of the 1000 LOESS regressions the 95 % confidence intervals are calculated and compared with the LOESS regression of the not combined individual set of profile observations. When the modelled LOESS regression of the original data set lies outside of the confidence interval, the null hypothesis, that there is no difference between the two original data sets, is rejected. For the LOESS regression we used all five profiles per subplot.

### 2.4.3 Statistical differences between median infiltration depths

To test for significant differences in observed infiltration depths among age classes and among irrigation intensities the non parametric Mood's median test was used (Hervé, 2018). The significance level was set to p < 0.05. The Mood's median test compares median pairs of two or more groups. A p-value lower than 0.05 indicates that at least the median of one group is





significantly different from the other groups. To identify which groups are statistically different a pairwise Mood's median tests across groups was used as a post hoc test (Mangiafico, 2016). The test was also used to detect significant differences between the dye pattern characteristics among the age classes.

To investigate differences and similarities in the relationship between PFF and site/vegetation characteristics, a linear regression

of PFF with site and vegetation characteristics was plotted by using the ggplot2-package (Wickham, 2016). A 95 % confidence interval was chosen for the significance test of the linear relationship. All data analyses were carried out using R (R Core Team, 2017).

## 3 Results

### 3.1 Vertical dye pattern analysis

We consider the median maximum staining depth as the representative infiltration depth of the soil profile. The representative infiltration depths are less than 1 m, but vary strongly between the age classes and irrigation intensities (Figure 3). Across the moraine ages the representative infiltration depths differ significantly (Figure 3, top), but do not show an age trend. A significant relationship between irrigation intensity and representative infiltration depth is only found at the 13.5ka, where the infiltration depth increases with irrigation intensity. Sorted by irrigation intensity, no distinct age trend in the infiltration depths

is observed (Figure 3, bottom).



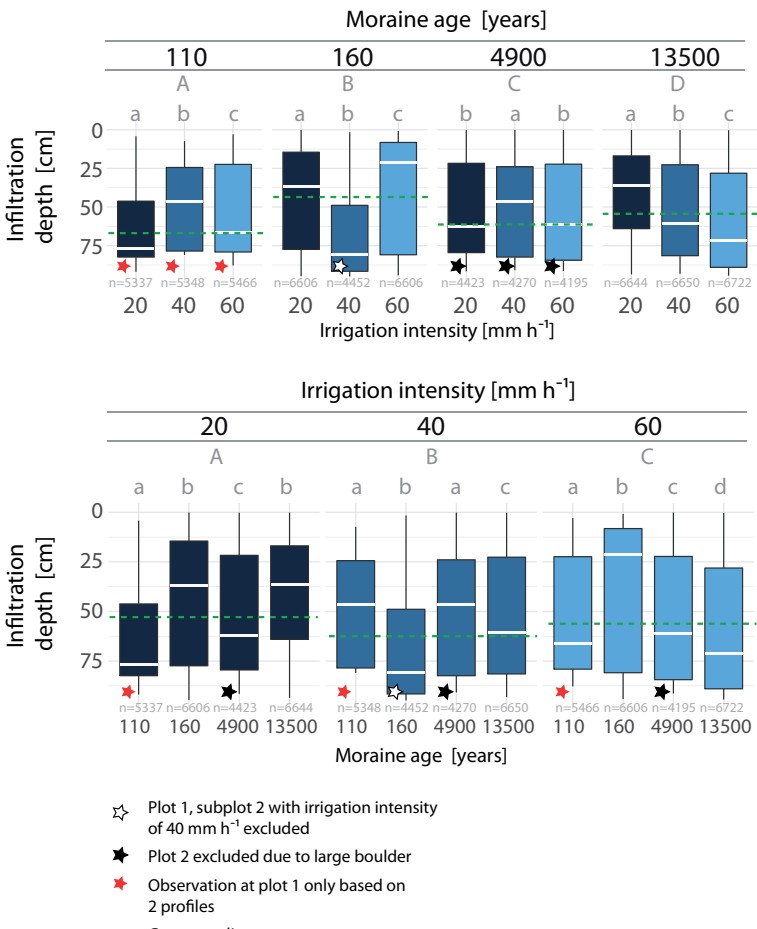

**Figure 3.** Maximum staining depth in each pixel column along the profile width at all excavated profiles at each age class (upper plot) and irrigation intensity (lower plot). Median values indicate the representative infiltration depth. Significant differences in median values are indicated by different letters. Upper case letters indicate the results of the Mood's median test in combination with a post hoc test among the age classes (upper plot) and irrigation intensities (lower plot). The dashed green line shows the group median used for the test. Different lower case letters denote the results among the irrigation intensity in each age class (upper plot) and the results among the age classes at each irrigation intensity (lower plot). n equals the number of pixel columns evaluated in each box plot.





**Figure 4.** Mean volume density profiles per age class and irrigation intensity. The volume density is the fraction of stained pixels, here color coded by flow path width (stained path width, SPW) and rocks. The modeled probability of a pixel being stained ($p_{dye}$) is shown in orange. Maximum infiltration depth was >1m in 31 of the 36 experiments. Arrows indicate the maximum infiltration depth < 1m at the remaining 5 plots.





The averaged volume density (VD) profiles (over the 5 profiles per subplot) of the three stained path width (SPW) classes per subplot are displayed in Figure 4. The sum of the VD-profiles of the three SPW classes per subplot is equal to the VD-profile of all stained areas (which is also equal to the dye coverage). At the two young moraines (110a and 160a) we observed clearer differences in the VD-profiles between the experimental plots of the single age groups than between the irrigation intensities

(Figure 4). At both young moraines the VD of SPW>200 mm is high over the entire profile depth at the plots labeled plot 3. These are also the two plots with a distinctly higher vegetation cover (Table 1) at these otherwise sparsely vegetated young moraines. At the two other plots of the 110a and 160a moraine, the fractions of SPW>200 mm are high in the upper 10-20 cm, but distinctly decline with depth. At the 4.9ka and 13.5ka the fraction of SPW>200 mm is lower in the upper 10-20 mm compared to the young moraines and over the entire profile range 20<SPW<200 mm predominantly have the largest share on the

dye coverage. The proportion of SPW<20 mm is negligible at all age classes.

Since the averaged VD-profiles (averaged dye coverage) could be affected by large individual deviations, we calculated the depth dependent probability of a pixel being stained ($p_{dye}(z)$) as a second way to describe the mean dye coverage profile per subplot (Figure 4). Comparing it with the averaged VD-profiles, we see that the course of the dye coverage profiles is well represented with this method. However, we also noted, that the high odds in the top 20 cm were mostly underestimated with

the linear model, since the measured log(odds) did not completely follow a linear course with depth (data not shown). Additionally, the logistic regression model only describes monotonously decreasing/increasing VD-profiles. It is thus not accurate for describing VD-profiles for systems in which VD varies greatly with depth, e.g. when large stones lead to a reduction of the available flow area in the middle part of the profile (as in 110a: plot 3, 4.9ka: plot 1 and 2, 13.5k: plot 1).

Most of the cross-section of plot 2 at 4.9ka and the subplot irrigated with 40 mm h$^{-1}$ of plot 1 at the 160a moraine was occu-

pied by a large boulder in the ground (see Figure A1). Please note that due to the strong inhibition of the water transport, these subplots were excluded from further analysis. Further, it must be taken into account that the results of plot 1 at 110a are based on only two observations (= two profiles per subplot), as the excavation had to be interrupted due to an unforeseen change in weather conditions. Despite protection, the rest of the experimental plot collapsed due to a thunderstorm.

The depth integrals of the dye pattern characteristics were compared across all experiments using boxplots, where each box

plot contained the information of 3 experiments à 5 soil profiles (Figure 5). The dye coverage (= integral of the VD-profile) corresponds to the area percentage of all blue-colored areas per profile (Figure 5 (a)). The proportions of the three SPW classes (Figure 5 (b-d)) are also given as the area percentage per profile. The dye coverage varies strongly (between the experimental plots) at the 110a and 160a moraine with the majority ranging between 10 and 60 % (Figure 5 (a)). The variation decreases with moraine age and the dye coverage at the oldest moraines mostly ranges between 20 and 50 %. The proportion of SPW<20 mm

increases slightly with age (Figure 5 (b)), but remains negligible with an average area proportion of less than 2 % on the total profile area. An increase with age can be seen in the area proportion of 20<SPW<200 mm (Figure 5 (c)) with a corresponding decrease in the area proportion of the SPW>200 mm (Figure 5 (d)).





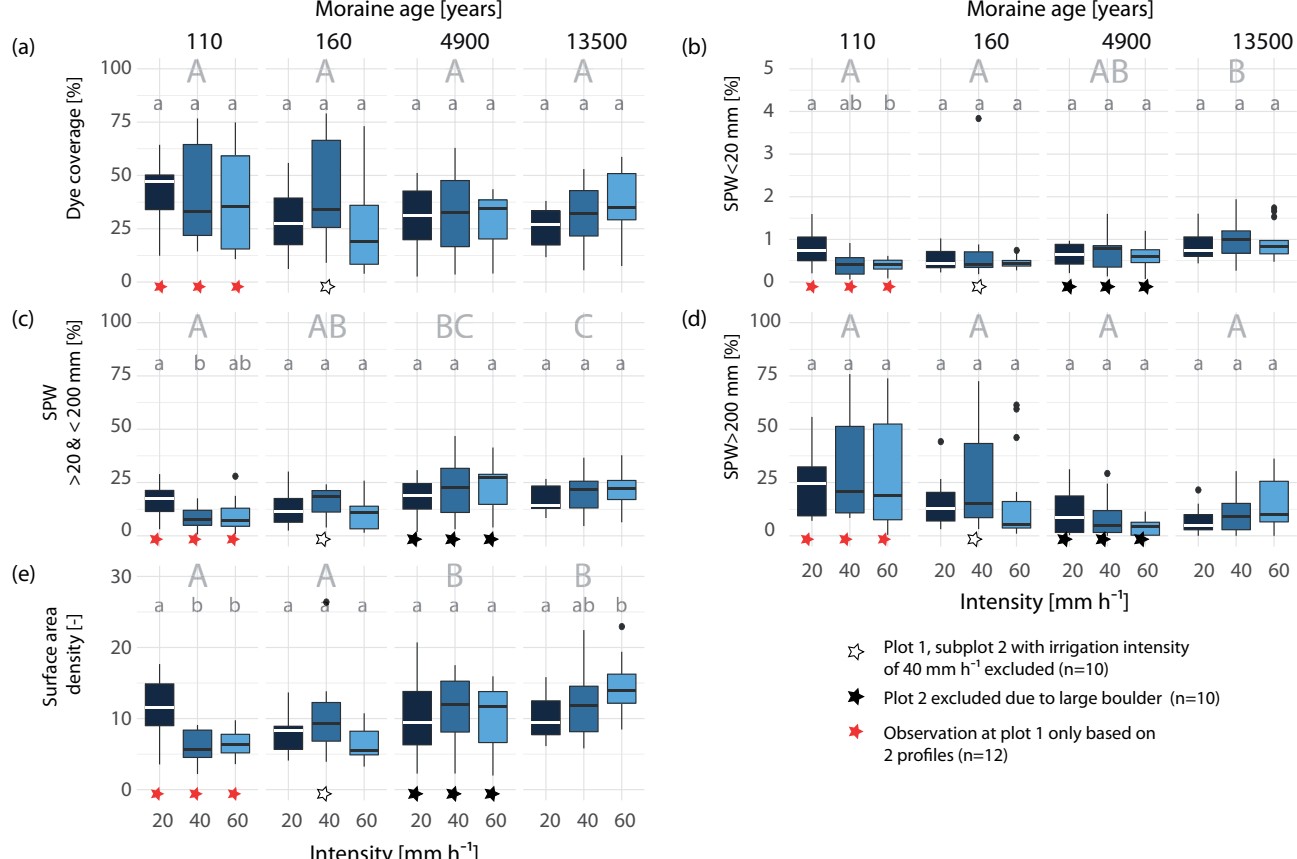

**Figure 5.** Box plots of dye staining characteristics compared across the three irrigation intensities and the four age classes (each box plot shows the data of 3 plots à 5 profiles, 15 profiles in total). (a) Dye coverage (volume density) in area percent of the entire soil profile, proportion of (b) stained path with (SPW) <20 mm, (c) 20 mm<SPW<200 mm, (d) and SPW>200 mm in area percent of the entire soil profile, and (e) integral of the surface area density (measure for the number of flow paths) per age class and irrigation intensity. Upper case letters indicate the results of the Mood's median test among the age classes. Different letters denote significantly different median values among the age classes (medians not shown). Lower case letters denote the results among the irrigation intensity in each age class.

To clarify if the stained area described by VD (Figure 5 (a-d) and Figure 4) is made up of many small flow paths or few large ones VD has to be jointly interpreted with the surface area density (SAD) (Figure 5 (e) and Figure 6), which is a measure for the number of individual flow paths. SAD increases along the chronosequence (Figure 5 (e)) with the exception of very high SAD values at the youngest moraine irrigated with 20 mm h$^{-1}$. A depth differentiated display of SAD (for the increments of 0-20, 20-40, 40-60, and 60-100 cm) per age class and irrigation intensity is given in Figure 6. At the young moraines, the SAD in the upper 10-20 cm is comparable to that of the old moraines, but decreases more strongly with depth (Figure 6). With increasing age, the decrease in SAD with depth is less and less pronounced. The combination of SAD and VD reveals distinct differences in the staining patterns along the chronosequence. The young moraines are dominated by a high VD (in some cases restricted





to the shallow depth only), a low SD, and a dominant fraction of SPW>200 mm. Whereas the old moraines have a high fraction of 20<SPW<200mm combined with a high SAD, which indicates a higher number of smaller, narrow blue-colored areas and thus more individual active flow paths at the older moraines and less individual flow paths but larger continuous areas used for water transport at the young moraines.

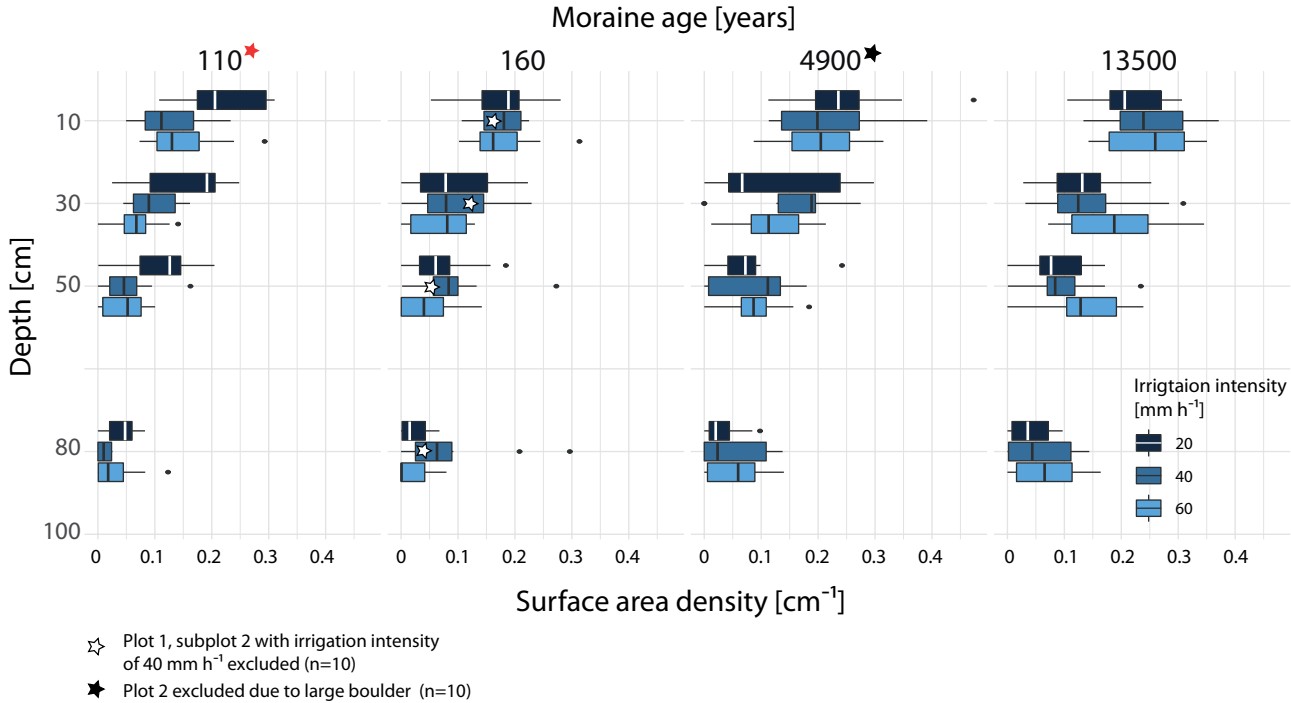

**Figure 6.** Surface area density (an indicator for the number of flow paths observed across the profile at a certain depth) for the depth increments of 0-20, 20-40, 40-60, and 60-100 cm per age class and irrigation intensity. Each box contains the information of the five profiles per subplot (n=15).

5  Different irrigation intensities mostly do not lead to significant differences between the resulting dye pattern characteristics (Figure 5). Even though there are no statistically significant differences between the medians in dye coverage, the medians at the young moraines tend to decrease and at the old moraine to increase with increasing intensity (Figure 5 (a)). The respective fractions of the three SPW classes also seem to be affected by irrigation intensity, but changes are only significant for the two SPW classes <200 mm at the 110a moraine. At this age class a reduction in the median areal fraction with increasing intensity is

10  observed for all three SPW classes. The interquartile range, however, increases for SPW> 200 mm. At the 4.9ka, a tendency to higher fractions of 20<SPW<200mm and a tendency to lower fractions of SPW>200mm with increasing intensity is observed. Whereas at the 13.5ka a tendency to higher proportions is found for both SPW classes. However, the differences between the intensity levels are not tested as statistically significant. In case of SAD trends differed between the age classes. A statistically





significant increase of SAD with irrigation intensity can be seen at the 13.5ka and a decrease at the 110a (Figure 5 (e)). The SAD at the 160a also tends to decrease with increasing intensity, but according to the Mood's median test the differences are not statistically significant, The depth distribution of SAD per irrigation intensity (Figure 6) shows an increase in SAD with increasing irrigation intensity at all depths at the oldest moraine and a tendency to a higher SAD at the 4.9ka starting at a depth

of 20 cm.

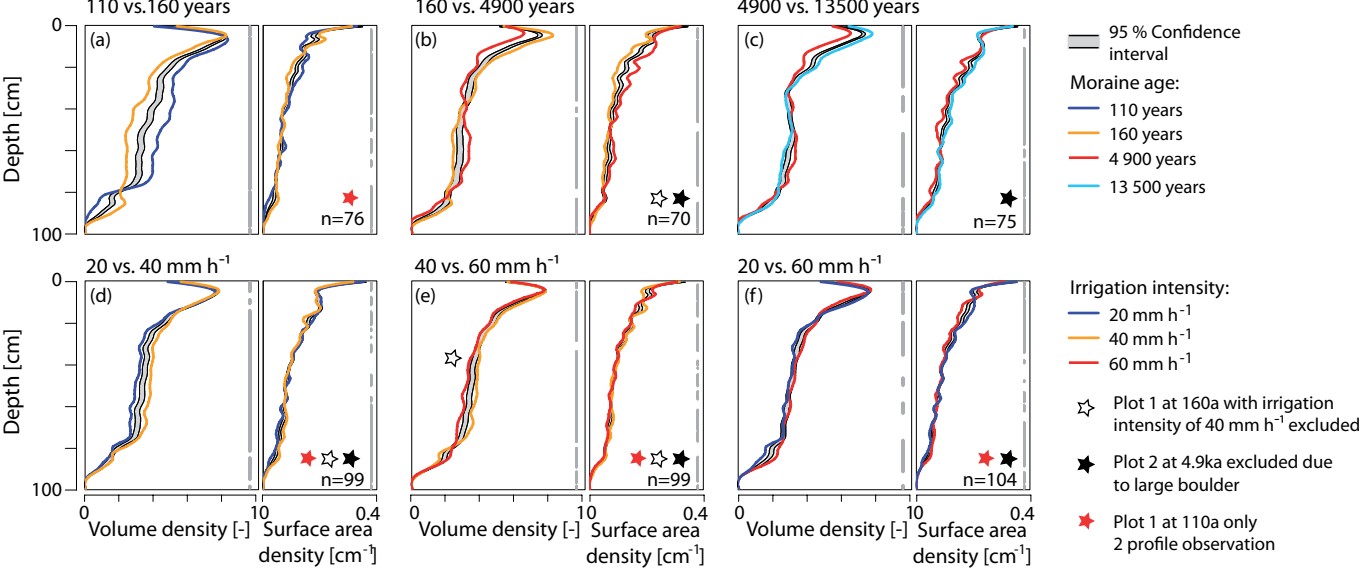

**Figure 7.** Differences of volume density and surface area density profiles with respect to (a-c) moraine age and (d-f) irrigation intensity. If the profile lines sit outside the grey-shaded confidence interval, the two profiles are considered to be significantly different. The parts of the depth profiles where this is the case are indicated by grey vertical bars on the right of each plot. n denotes the number of profiles used for the depth-wise re-sampling.

To quantitatively asses the impact of age and irrigation intensity on the dye pattern a statistical approach in form of a bootstrapped LOESS regression (BLR) was used. The approach is designed for a comparison of two data sets with profile observations (Keith et al., 2016). The results of the BLR approach for a pair wise comparison of the averaged volume density

and surface area density profiles are shown in Figure 7. Next to the 95 % confidence interval of the 1000 LOESS regressions (bootstrap resampled out of the combination of both compared data sets) the LOESS regression of both original data sets are shown. The differences between the two profiles are significant if the LOESS regression curves sit outside the confidence interval. It would actually be sufficient to plot only one LOESS regression of the original data sets (Keith et al., 2016), but we plotted both. Based on the moraine age as the test variable we compared the sets of all volume density and surface area

profiles per moraine age along the chronosequence (Figure 7 a-c). We find statistically significant differences in the volume density and surface area density profiles among the neighboring age classes. The grey vertical bar indicating where differences





are significant is almost continuous and has only a few short interruptions.

**Figure 8.** BLR-test for differences in volume density profiles and surface area density profiles among the three irrigation intensities per age class. If the profile lines sit outside the grey-shaded confidence interval, the two profiles are considered to be significantly different. The parts of the depth profiles where this is the case are indicated by gray vertical bars on the right of each plot. n denotes the number of profiles used for the depth-wise re-sampling.







**Figure 9.** BLR-test for differences in volume density profiles and surface area density profiles among the three experimental plots per age class. If the profile lines sit outside the grey-shaded confidence interval, the two profiles are considered to be significantly different. The parts of the depth profiles where this is the case are indicated by gray vertical bars on the right of each plot. n denotes the number of profiles used for the depth-wise re-sampling.

When comparing the profiles with regard to the irrigation intensity irrespective of age, we see that significant differences are dominating, but the LOESS-regression profiles are mostly located very close to the confidence interval (Figure 7 d-f). The profile comparison between irrigation intensities carried out individually for each age class, shows similar results (Figure 8). However, especially at 110a the VD-profiles do not differ significantly in the middle part of the soil profile. This is true for

5  all comparisons among the irrigation intensities. At the 4.9ka moraine the gray bars indicating significant differences are often disrupted with a slight tendency towards more significant differences in the upper half meter for VD. In contrast at the 160a





moraine and especially at the 13.5ka moraine, the regression lines are located far outside of the confidence interval, indicating significant differences across all irrigation intensities (except 20 vs. 60 mm h$^{-1}$ at 160a).

To investigate the spatial variability within each age class we compared the profiles of the three plots per moraine (Figure 9). In this case profiles across all irrigation intensities were used, revealing significant differences between the three plots for all

age classes. The profiles at the 4.9ka and 13.5ka moraine show mainly significant differences, with no comparison within the age class being particularly striking. The LOESS-regression profile lines comparing plot 1 and 2 with plot 3 at the two young moraines (110a and 160a), however, lie far outside the confidence interval, which indicates strong differences between the profiles at plot 3 and the other two plots. This resembles the strong differences in the mean VD-profiles (Figure 4) at the two young moraines. Plot 3 at both moraines, clearly stands out from the other plots with a higher vegetation cover and a larger VD

at greater depths.

## 3.2 Flow type classification

Using the VD-profiles of the three SPW classes and their proportion on the total dye coverage to characterize flow types (Weiler, 2001) we found that over the millennia flow types transition from matrix flow to preferential flow in form of finger flow (Figure 10 a). At the youngest moraine matrix flow is the predominant flow type (relative frequency > 0.6) followed

by the flow type class 'Macropore flow with high interaction/ Finger flow'. The dominant flow type of this joint flow type class (Hartmann et al., 2020a) is finger flow along narrow flow paths, since no actual macropore flow was found during the field experiment. At the 160a the relative frequency of matrix flow already decreased to 0.5 and the frequency of finger flow increased. At the two oldest moraines the dominant flow type is finger flow and the relative frequency of matrix flow dropped below 0.3. Considering the entire profile depth of 1 m, the frequency of matrix flow decreases and the frequency of finger flow

increase continuously with moraine age. A depth differentiated view shows a higher proportion of finger flow at the 4.9ka than at the 13.5ka in the upper 20 cm (Figure 10 a1). In the other depths (Figure 10 a2 to a4), however, a continuous increase in finger flow frequency with moraine age was observed.

With regard to the irrigation intensity no consistent impact on the flow type distribution across the millennia could be identified (Figure 10 b). At the 110a and 160a moraine the two dominant flow types (matrix flow and finger flow) show an almost equal

distribution across all irrigation intensities. A tendency to less matrix flow is observed at the 4.9ka, whereas at the 13.5ka the frequency of matrix flow increases with increasing irrigation intensity. Differentiated by depth, we observed no systematic trend in flow type frequency distribution with increasing irrigation intensity in the upper 20 cm for all age groups (Figure 10 b1). From a depth of 20 cm, the 4.9ka and 13.5ka each show a trend-like behavior in the shift of the frequency distribution with irrigation intensity analogous to the observation of the entire soil profile (Figure 10 b2 to b4). From a depth of 40 cm, the

relative frequency of matrix flow also increases with increasing irrigation intensity at the 110a and 160a (Figure 10 b3 to b4).

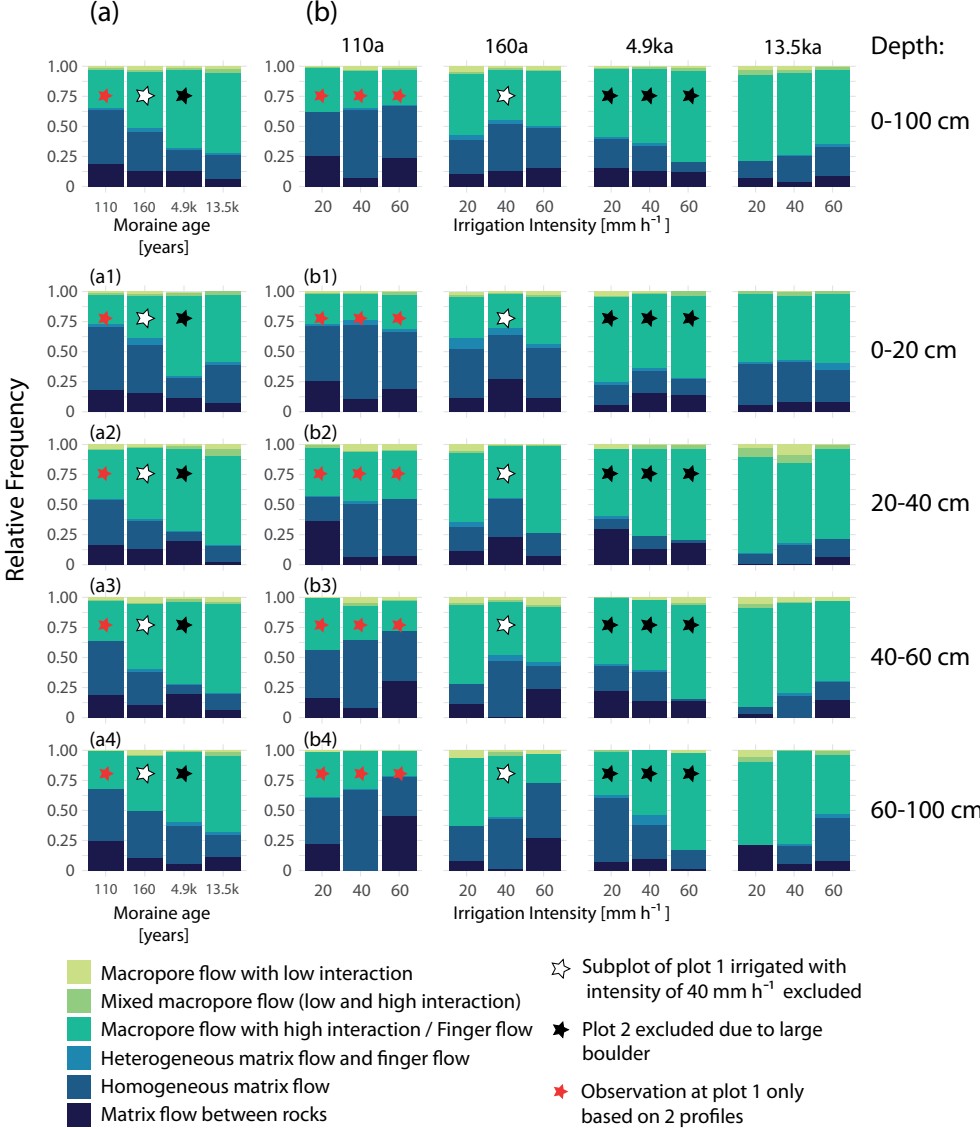

**Figure 10.** Relative frequency distribution of flow types (a) for the four moraine age classes, and (b) differentiated by irrigation intensity for each moraine age. The flow type frequency distribution is displayed once for the entire profile depth of 100 cm (a and b) and once differentiated by four depth segments (a1 to a4 and b1 to b4). Matrix flow types are displayed in a blue color scale and preferential flow types in a green color scale.

## 3.3 Correlation of preferential flow frequency with site characteristics

Infiltration patterns and the formation of subsurface flow paths can be related to site characteristics. We tested the correlation between the preferential flow fraction (PFF) in the topsoil (0-20 cm) and the site characteristics listed in Table 1 by applying

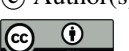



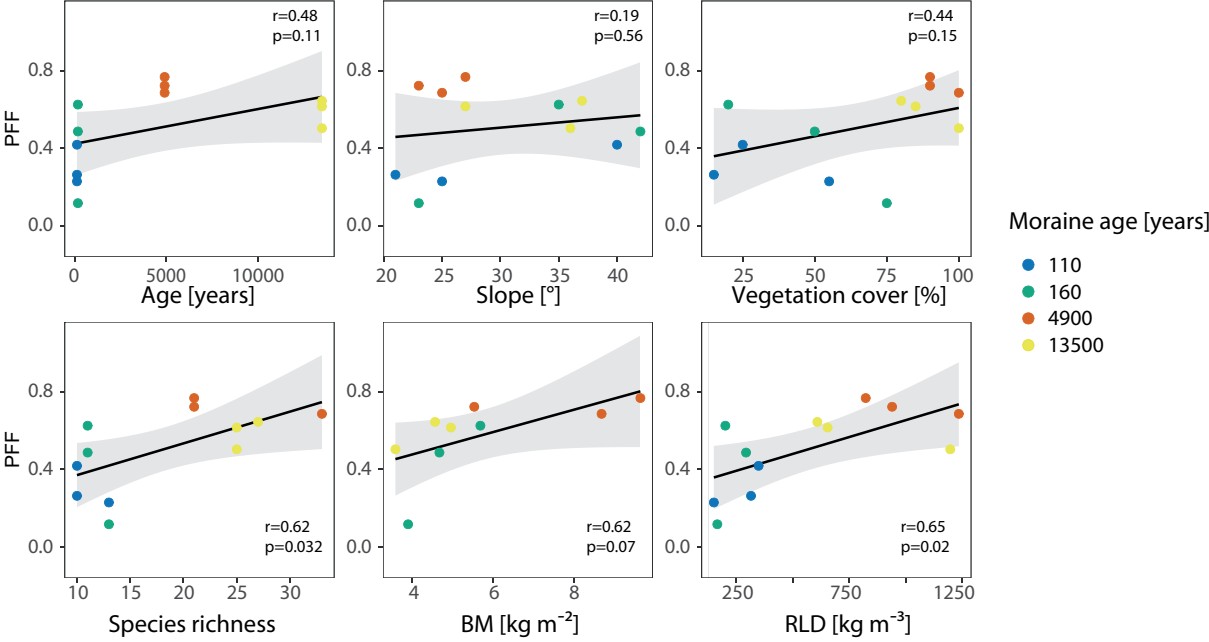

**Figure 11.** Frequency of preferential flow (PFF) in the upper 20 soil centimeters in relation to moraine age, slope, and vegetation characteristics (BM= above ground biomass, RLD= root length density).

a simple linear model (Figure 11). These site characteristics include age (which summarizes many physical and biotic characteristics), slope as a purely physical descriptor and then several vegetation characteristics, such as vegetation cover, species richness, above ground biomass, and root length density. Although the sample size is quite small and the analysis cannot identify direct causalities and does not account for possible multi-collinearities, it nevertheless provides some insight on potential

5   controls of preferential flow occurrence. Between moraine age and PFF we only observed a small and at the 0.05-level statistically not significant (p>0.05 at, r<0.5) correlation. A correlation between slope and preferential flow occurrence can also be ruled out (r<0.2). The vegetation cover shows the weakest correlation with PFF (r<0.5) out of the four tested vegetation parameters, whereas the species richness, BM, and RLD have a stronger correlation with PFF (r>0.6). The relationship (r>0.6) between PFF and BM is not statistically significant (p=0.07). However, since the p-value is affected by sample size, we have

10   to point out that due to missing data at the 110a moraine, the sample size for BM is reduced compared to the other vegetation properties (n=9 instead of n=12). This can negatively affect the comparability of the correlations. The relationships of PFF with species richness and RLD are both statistically significant (p <0.05), whereas the strongest linear relation exists between PFF and RLD (r=0.65).



## 4  Discussion

### 4.1  Evolution of flow paths

Across the chronosequence we observed significant differences in the dye patterns (Figure 7). Based on the flow type classification of the dye patterns we found that the frequency of matrix flow decreases with age and the frequency of finger flow

increases. The frequency of matrix flow is especially high (>0.6) in the top soil (0-20 cm) at the 110a and 160a moraine. However, at these two moraines, distinctly more surface runoff was observed during the irrigation experiments than at the older moraines. At both young moraines, the high amounts of surface runoff (unfortunately unquantified) were mainly observed at plot 1 and 2. From purely visual observation during irrigation it seemed like the amount of surface runoff increased with irrigation intensity at the younger moraines. At these four plots the water infiltrated homogeneously in the top 10 cm, but there

was hardly any staining in the soil below this depth (Figure 4). Lateral subsurface flow, however, was not observed. In contrast at plot 3 at both young moraines deep infiltration and vertical homogeneous water transport was observed.

We link the differences in infiltration patterns and resulting staining patterns to the differences in vegetation cover. Whereas both plots labeled as plot 3 showed a high degree of vegetation coverage (> 50 %, Table 1), which was evenly distributed over the entire plot area (Figure 12), plot 1 and 2 at both moraines had a low vegetation cover with only a few single vegetation

patches between gravel and small stones. We hypothesize that structural sealing could be the cause for the reduced infiltration depths and higher amounts of surface runoff at plot 1 and 2 of the two young moraines, as surface runoff started only a few minutes after the start of the irrigation. At both moraines, the aggregate stability of the loamy to sandy soils with low organic matter content was found to be comparatively low (Greinwald et al., 2021c). Thus, high irrigation intensities are suspected to have caused the structural sealing of the soil surface, which induced overland flow on these otherwise coarse textured soils with

high saturated hydraulic conductivities (Maier et al., 2021) and a small water holding capacity (Hartmann et al., 2020b). The disruption of the soil surface structure due to irrigation and the wash-in of released fine particles can lead to clogging of near-surface pores (Assouline, 2004) which results in a reduction in near-surface porosity and unsaturated hydraulic conductivity (Armenise et al., 2018). This phenomenon was also suggested by Maier and van Meerveld (2021) during large-scale sprinkling experiments at the same moraines. The high vegetation cover at the other plots likely protected the soil surface from the impact

of the irrigation and a homogeneous and deep infiltration was observed (Figure 4).

The pore space of coarse-textured, unsorted, sandy soil is mainly made out of large pores, which provide only a low water retention capacity and lead mostly to a fast downward transport of water (Hartmann et al., 2020b). The root system has a low density (Table 1, Greinwald et al. (2021a)) at the early stages of vegetation succession and does not impact the water transport. The sparse vegetation cover also does not inhibit infiltration and the water can infiltrate deep into the soil. Only larger

stones and occasional clay lenses (the size of a few centimeters) or other material heterogeneities influence the water transport and create heterogeneous matrix flow. At the two oldest moraines, the vegetation cover was dense and probably has a high interception storage capacity (albeit reduced by vegetation trimming). The fine textured soil with a higher porosity (and higher proportion of fine pores) and lower bulk density has a higher water retention capacity than the two young moraines. The soil is heterogeneous with a higher organic matter content in the upper layer and depth-gradients in porosity (decreasing) and bulk





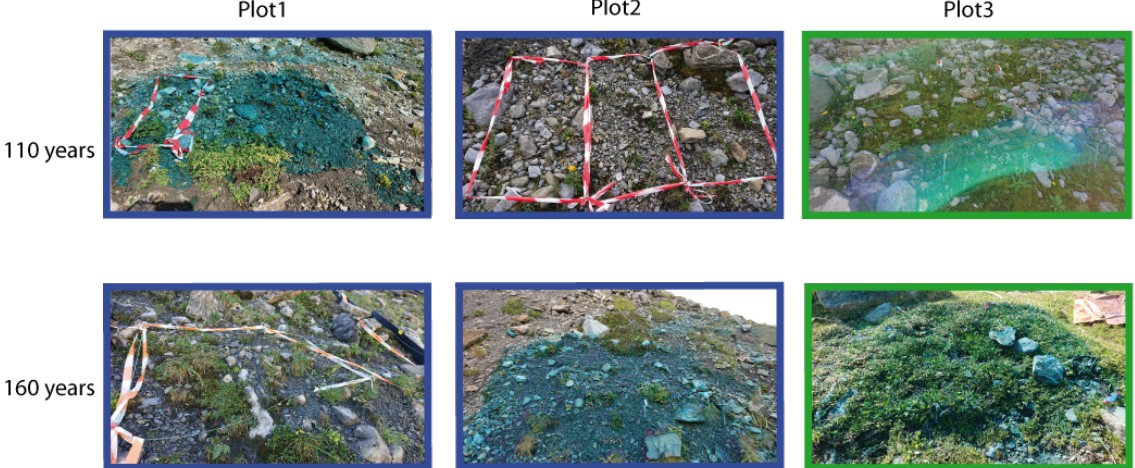

**Figure 12.** Surface of the plots at the 110 and 160-year old moraine. Frame color indicates plots with a similar degree of vegetation coverage. Photograph of the high complexity plot at 110-year old moraine is disrupted by lens flares due to backlighting.

density (increasing) (Hartmann et al., 2020b). The root system is dense with most of the root mass (> 90 %) located in the uppermost 30 cm (Greinwald et al., 2021a). At both moraines deep infiltration, but almost no surface runoff no subsurface lateral flow was observed. The image analysis has shown that finger flow dominates (Figure 10), with fingers already induced at the soil surface or within the upper 20 cm. Thus, water infiltrated heterogeneously and/or water transport was affected by

properties of the soil surface or of the upper soil layer.

Heterogeneous infiltration patterns under grass cover causing finger flow with large parts of dry soil were also observed by de Jonge et al. (2009), who found water repellency to be the main cause for this flow pattern. The hydrophobicity index (HI) at each moraine was measured by (Maier and van Meerveld, 2021). The mean HI at the youngest moraine is small (HI<2) and increases continuously with age (4.9ka: mean HI=5.05; 13.5ka: mean HI=9.36). The fraction of preferential flow paths in the

top 20 cm, however, was highest at the 4.9ka moraine (Figure 10). Inaccuracies in the comparison of HI with our results can arise from the dependency of soil hydrophobicity on soil moisture (de Jonge et al., 2009) and the fact, that HI was measured outside the experimental plots on different days than the irrigation experiments (likely different antecedent soil moisture). Water repellency is also positively correlated with the organic matter content (de Jonge et al. (2009), Mataix-Solera and Doerr (2004)) and is also higher when the organic matter is made up of complex compounds (Mainwaring et al., 2004)).

Hartmann et al. (2020b) found a slightly higher organic matter content in the top 20 cm at the 4.9ka compared to the 13.5ka, which corresponds with the higher fraction of finger flow at the 4.9ka. Hydrophobic organic compounds are also released by root activities (Doerr et al., 1998). We observed a significant correlation between the increase in preferential flow paths in the upper 20 cm and root length density and the above ground biomass. The connection of preferential flow paths and RLD is probably also attributed to the fact that roots form channels which improve infiltration (Zhang et al., 2015). The strong corre-

lation with above ground biomass, but less correlation with vegetation cover is more difficult to explain and likely depends on





the vegetation species. The vegetation cover at the 13.5ka was mainly composed of dense grass vegetation, while at the 4.9ka also shrub was occasionally present, which on the one hand produces more biomass, but also forms a root system with a higher quantity of larger roots, which enhances infiltration locally.

Soil layering with layers of fine texture above coarse texture are known to facilitate the formation of finger flow (Morales
et al., 2010). In such layered soils, homogeneous infiltration fronts become unstable and break into finger flow at the material boundaries (Starr et al. (1978), Hendrickx and Flury (2001), Wang et al. (2018)). At our sites, weathering and organic matter accumulation formed a surface layer with a finer grain size and higher water retention than the unweathered and coarser soil below, which is clearly pronounced at the 4.9ka and 13.5ka moraine (Hartmann et al., 2020b). Also observed material heterogeneities such as gravel and sand patches likely facilitated the formation of finger flow paths.

Despite the almost 10000 year age difference between the 13.5ka and the 4.9ka moraine, the 4.9ka shows a higher proportion of preferential flow paths in the upper 20 cm, which is also associated with a higher proportion of BM, RLD, and vegetation cover. In addition, the clay content, organic matter and porosity are also higher at the 4.9ka moraine (Hartmann et al., 2020b). This discontinuous trend raises concerns that the two moraines do not fall within the chronosequence approach assumptions that time is the only variable during landscape development. Thus it cannot be excluded that e.g. different initial site conditions,
climate boundary conditions, or/and geomorphological disturbances could have led to different rates of change (Wojcik et al., 2021) in the topsoil.

Compared to the results decribed by Hartmann et al. (2020a) for soils developed from siliceous glacial till under similar climatic conditions, we found distinct differences in the flow path evolution at this calcareous forefield. While the flow path development was nearly identical in the first 5000 years in both geologies, it differs distinctly after ten thousand years of land-
scape development. In both geologies the flow paths developed from a more or less homogeneous to heterogeneous matrix flow at 160 years to finger flow after 5000 years. After more than 10000 years of landscape development, subsurface hydrology at the calcareous geology is ruled by finger flow and deep infiltration, whereas a the siliceous geology storage capacity in the top soil strongly increased with a corresponding reduction in infiltration depths and a shift to macropore flow.

## 4.2 Impact of irrigation intensity

Studying the impact of irrigation intensities on subsurface flow paths is often hampered by the influence of different initial and boundary conditions (e.g. Wu et al. (2015), Cichota et al. (2016)). Our study was specifically designed to minimize these effects by dividing the irrigation plots into three adjacent subplots. This allows for the assumption that the initial and boundary conditions (excluding the controlled irrigation intensity) of the subplots per plot were almost identical. From the end of July to the beginning of September 2019 we measured a precipitation amount of 360 mm at the glacier forefield. Matrix potentials,
measured by tensiometers in 10, 30 and 50 cm at the 160a, 4.9ka and 13.5ka, never dropped below field capacity. Thus, the initial soil moisture was consistently high across the irrigation experiments at all age classes.

The observed changes in dye patterns and flow paths with increasing irrigation intensity at the four age classes are schematically summarized in Figure 13. At the 13.5ka we observed an increase in dye coverage (Figure 5 (a)), an increase in infiltration depth (Figure 3), a broadening of the flow paths (Figure 5 (c-d)), and an increase in the number of flow paths (Figure 5 (e)





and Figure 6) with increasing irrigation intensity. Based on the increasing proportion of SPW>200 mm on the dye coverage the flow type classification indicates a transition to more matrix flow (Figure 10). At the 4.9ka moraine, we also observed an increase in dye coverage (Figure 5 (a)) and an increase in the number of flow paths (Figure 5 (e)). But other than at the 13.5ka, the proportion of 20<SPW<200 mm increases (Figure 5 (c)) and the proportion of SPW>200 mm decreases (Figure 5 (d)),

which then leads to an increase in the frequency of finger flow paths in the flow type classification (Figure 10 (b2) to (b4)). However, the process of flow type classification is only based on the proportions of the three SPW classes on the dye coverage (Weiler, 2001). The number of flow paths or the dye coverage itself are not taken into account. We observed at both age classes that with increasing irrigation intensity more fingers are generated and more soil space is used for water transport (Figure 5). It also has to be stated that at sites where preferential flow occurs in form of macropore flow the relations can be different, since

the controls inducing macropore flow are different (Nimmo, 2021).

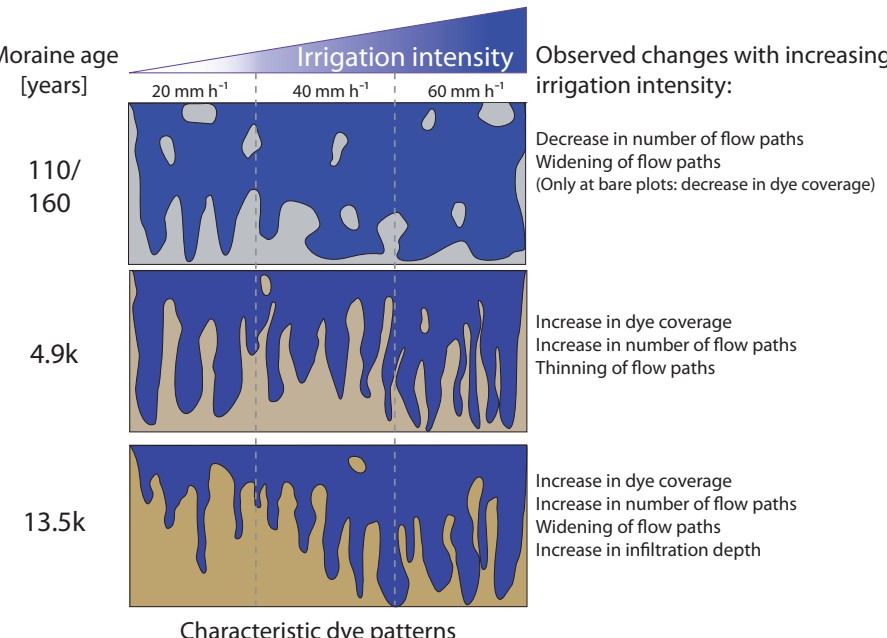

**Figure 13.** Characteristic dye patterns and observed changes with increasing irrigation intensity at the four age classes.

The formation of finger flow paths and their properties such as number, flow velocity, or width are in a complex interplay with the surrounding soil moisture, the flux rate and soil properties (Nimmo, 2021). Studies focusing on the formation of finger flow paths found the finger width not only to be influenced by soil properties, and initial and boundary conditions (Glass et al., 1989), but also by the flow rate through the finger (Parlange and Hill (1976), White et al. (1976)) with higher flow rates

leading to an increase in finger width. This was also observed by Ma et al. (2008), who also found a positive correlation between rainfall intensity, time of finger flow occurrence and mean velocity. The increase in mean velocity of the fingers leads to a faster





downward transport and thus deeper infiltration depths with higher irrigation intensities (Cremer et al., 2017). An increase in the number of fingers with higher flux rates was also observed (Sililo and Tellam, 2000). These findings by other studies are similar to our observations at the 13.5ka moraine. It is unclear what causes the different observations in the dominant flow path widths at the 4.9ka and 13.5ka moraines. We can only speculate whether the higher organic matter content, the higher root

density, or soil properties such as the lower hydraulic conductivities and higher porosity play a role in producing narrower flow paths with increasing irrigation intensity at the 4.9ka moraine.

At the young moraines, however, we observed an in increase in the frequency of matrix flow at greater depths with increasing irrigation intensity, which is caused by an increase in the SPW>200 mm with a simultaneous decrease in the median dye coverage and a decrease in the number of flow paths (SAD) (Figure 5e). The decrease in median dye coverage with increasing

intensity is particularly pronounced at the bare plots (data not shown). No clear trend can be seen at the plots with a higher vegetation cover. However, the decrease in SAD and the increase in stained path widths indicates that water flow paths that reach greater depths tend to widen and to merge together with increasing irrigation intensity. This process might be facilitated by higher water contents at greater depths or by a change in material properties.

### 4.3  Uncertainties

Apart from the reduced sample size at plot 1 at the 110a moraine and the uncertainties due to large boulders at the 160a and 4.9ka moraines, some further uncertainties need to be mentioned. In general, the dark gray soil color at the 110a and 160a moraine made the color detection of the tracer difficult. In addition, at plot 3 of the 110a moraine, a set up of suitable lighting conditions was difficult due to stormy weather conditions. As a result, the lighting of the photographs was very unfavorable for the image analysis. Even during the profile excavation in the field it was not possible to determine with great certainty whether

the dark colored, wet soil was stained or not. Thus, blue stains on larger stones along the profile depth were considered as an indicator for the validity of the observed dye tracer pattern, which shows an almost complete coloring of the soil (Figure 4).

We further assume that the irrigation with the hand-operated sprayer, which had to be guided close to the soil surface due to strong winds, in part led to a high force of application and promoted structural sealing at the barer plots of the 110a and 160a moraines. At both moraines, deep infiltration was often found at the boundaries of the plots that were less affected

by the direct application of water. This observation suggests that a more homogeneous and deep transport of the water can take place in this quite homogeneous and unsorted material (Hartmann et al., 2020b), if the surface is not influenced by particle displacement. Thus, it is assumed that the proportion of preferential flow paths at the young moraines is generally overestimated and homogeneous to heterogeneous matrix flow with deep infiltration are the dominant flow types under natural rainfall conditions. As the plot boundaries are excluded from the image analysis to avoid edge effects, the here observed deep

percolation could not be accounted for in our qunatitative analysis.



## 5 Conclusions

Based on Brilliant Blue dye experiments in a glacial chronosequence on calcareous parent material, we found that subsurface flow paths change with age: from homogeneous gravity driven matrix flow in sandy, coarse-grained, loose soils with low root density at the young moraines to a heterogeneous matrix and finger flow in silty, layered soils with dense vegetation cover

and high root density at the old moraines. The occurrence of preferential flow paths increases with soil age, but as it appears only in form of finger flow, it is mainly controlled by soil surface characteristics (organic matter content, soil texture, soil layering) and vegetation characteristics (RLD, hydrophobicity, BM). When infiltration was not impaired by structural sealing, water percolated deep into the soil (> 1 m) at all four age classes. The observed finger flow paths and deep drainage even after more than 10000 years of landscape development contrasts with the observed high storage capacity, reduced infiltration, and

occasional macropore flow in a moraine on siliceous parent material of the same age (Hartmann et al., 2020a). The observed differences in flow path evolution in these two different geologies under nearly identical climate conditions emphasizes the important role of the parent material in landscape evolution. We also found an increase in the number of preferential flow paths (finger flow paths) with increasing irrigation intensity at the two old moraines, which leads to an increase in soil space used for water transport and thus efficient infiltration preventing surface runoff even at high intensities.

Our findings deliver important insights on how landscape evolution affects hydrological processes in transient alpine land-scapes, where glacial retreat is accelerating and thus more and more hillslopes are freed of ice, and weathering, erosion and plant succession are initiated. The here provided data and observations can also help to improve the handling of hydrologic processes and their role within the feedback cycle of the hydro-pedo-geomorphological system when it comes to soil and landscape evolution modeling.

*Code and data availability.* The soil structure and texture data is described in detail in Hartmann et al. (2020b). It is available at the online repository of the German Research Centre for Geosciences (GFZ, Hartmann et al. (2020c)) and can be accessed via the DOI: https://doi.org/10.5880/GFZ.4.4.2020.004

The Brilliant Blue images can be obtained from the authors upon request.



# Appendix A

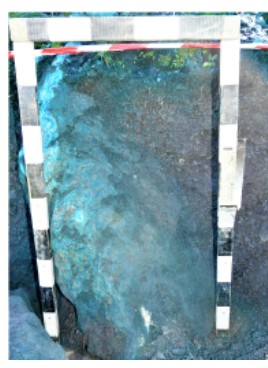 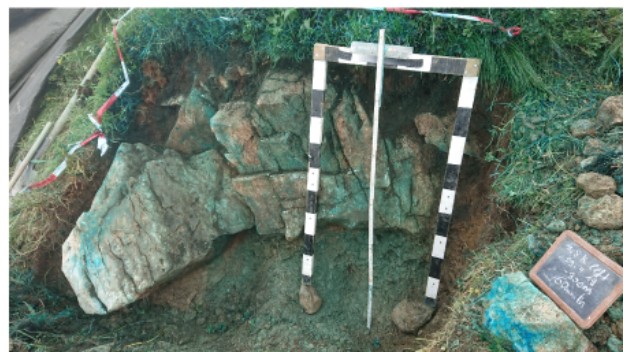

**Figure A1.** Hidden boulders at the 160a (left) and 4.9ka (right) moraine. The boulder occupied most of the cross section of the subplot irrigated with 40 mm h$^{-1}$ at plot 1 of the 160a and most of the cross section of plot 2 at the 4.9ka moraine. Due to this strong disturbance, the corresponding plot and subplot were neglected in the further analyses. The length of one black/white scale segment of the wooden frame equals 10 cm.

*Author contributions.* AH conducted the tracer experiments, soil sampling, and the laboratory analysis. KG provided information on site and vegetation characteristics. AH prepared the images and performed the analyses in discussion with TB. MW and TB were involved in planning the fieldwork. AH prepared the manuscript with contributions from all co-authors.

5 *Competing interests.* The authors have the following competing interests: Markus Weiler is editor and Theresa Blume is chief-executive editor of HESS.

*Acknowledgements.* This work is funded by the German Research Foundation (DFG) and the Swiss National Science Foundation (SNF) within the DFG-SNF-project Hillscape (Hillslope Chronosequence and Process Evolution). We thank Nina Zahn, Wibke Richter, Louisa Ka-nis, Peter Grosse and Carlo Seehaus for their persevering assistance in the field. We also thank the Canton Uri, the community Unterschächen
10 and Korporation Uri for permission to conduct the experiments. Many thanks to Christine, Franz and Matthias Stadler at Chammlialp for their support and kind hospitality.



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
