# Peer review of "Subsurface flow paths in a chronosequence of calcareous soils: impact of soil age and rainfall intensities on preferential flow occurrence"

_Hydrology and Earth System Sciences, 2022_

## Author Comment (AC1)

**Response to Reviewer comments**

**Response to Reviewer 1**

General Comments
Hartmann et al. present work on infiltration experiments across a moraine chronosequence in the Swiss alps, spanning almost 14,000 years. They performed infiltration experiments on four plots of different ages, with each plot being subdivided into three subplots where different precipitation intensities were applied.

The work is a heavily revised version of a previously submitted manuscript. I was one of the original reviewers back then and suggested a rejection. This submission was deemed different enough to be considered a new submission, and I would agree with this assessment. The way the authors reworked the current manuscript makes it much more enticing and sets it apart further from Hartmann 2020a and 2020b (in my opinion).

The authors would like to thank the reviewer for spending her/his time on reviewing our revised manuscript. We are pleased that the changes to the original manuscript were well received and appreciate the feedback and interesting comments.

In the introduction, the authors describe the need for identifying different flow patterns for potential integration into landscape evolution models. It would be interesting to revisit this idea in the discussion. What does the work suggest such an integration could look like, and more importantly, what would the practical differences be between the different flow types for such a model (especially considering that some of the differences between the plots appear minor, even though they were found to be significant)?

This is an interesting point, which we will include in the discussion of the revised manuscript.
First of all, we would like to state that we cannot derive any generally valid statements on the basis of the observations from this manuscript and Hartmann et al., 2020, since the observations are primarily only valid for the landscape development based on loose moraine material in siliceous and calcareous glacier forefields. Since our study made it clear that, due to the complexity of landscape development, time should not be regarded as the only primary evolutionary factor, it would be necessary to intensify correlation studies, such as those already vaguely tackled in Figure 11, in order to integrate a corresponding modelling approach in SLEMs. The aim of this study would be to identify the main influencing factors (e.g. soil properties, vegetation characteristics) on the formation of preferential flow paths and to create a kind of transfer function between properties and degree of preferentiality. The derived preferential flow frequency index (PFF) could be a helpful variable to quantify the extent of preferential water transport. The minor differences in the profiles between the plots would therefore not be of great importance.
How the final exact implementation of the description of the flow types in the model could look like largely depends on the desired complexity/accuracy of the target model (empirical/physically based).

I am also wondering if the amount of rocks has an effect on the flow type. If a large fraction of the profile is taken up by rocks, percolating water will be restricted to the space between the rocks. The authors do include flow types that take into account rocks in the profile. From what I understand, though, this applies mostly to homogeneous flow that happens around the rocks. What would happen if a larger rock led to an effective partitioning of an otherwise homogeneous wetting front? (That is, if the soil below the rock remained dry)

In the case of large rocks leading to partitioning / funneling of the wetting front, this funnel flow is automatically counted as finger-like flow within the image analysis (when the redirection of water leads to thinner flow paths). We will clarify in the revised manuscript that the flow type class finger flow refers to the water transport pattern and not specifically to the narrow sense of fingers created only by instabilities of the wetting front. Since we are primarily concerned with what the flow patterns look like and what they mean for water transport and not with a precise identification of the mechanisms behind the flow patterns, both flow types are classified in the same class of flow patterns. Both flow types lead to a fast and preferential vertical downward transport of the water by creating the same flow pattern.

Further, it appears that some profiles exhibited a significant portion of rocks in the upper soil layers. Overland flow was not measured, but it could be beneficial to talk a little more about the potential impact of less water infiltrating at these sites.
In the discussion, the authors mention very briefly that the edges of the plots were not analyzed. I might have missed this earlier, but does this only apply to the outer edges of the 1x1.5 m plots or also to the borders between plots 1 and 2 and plots 2 and 3? If so, how big of a buffer was included? I could imagine that interactions around the inner boundaries could have an impact, too.

To avoid a possible impact by interactions around the inner boundaries we excluded a soil space between neighboring subplots with a width of at least 6 cm from the analysis. We will include this information in the method section in the revised manuscript.

I think the revisions are a little more than just minor, but I am confident that the authors can address them.

Page 4, Lines 9-11: What is the reasoning for having two plots of ~the same age?

The original reason for selecting the 110 year old moraine as the youngest and the 160 year old as the second youngest moraine was mainly the result of local conditions at the glacier forefield. The actual goal was to select age groups that were as identical as possible to the moraine ages selected in Hartmann et al., 2020a (30a, 160a, 3000a, 10000a). This was not entirely possible for the youngest moraine. The choice of the 110 year old moraine as the youngest moraine is the result of the local conditions, as no adequate moraine with an age of around 30 years could be identified that also ensured comparability in terms of elevation and microclimate. We therefore had to compromise and selected the moraine with an age of 110 years as our youngest moraine (Musso et al., 2019).

Page 7, Lines 24+ Can you describe what the practical differences are between the flow types?

We will include the following explanation in the revised manuscript:
(1) Macropore flow with low interaction: water flow in macropores, where a low permeable or saturated soil matrix leads mainly to water flow within the macropore and only small lateral interactions between the macropore water and the soil matrix occur. The characteristic dye pattern shows narrow but long individual stains.
(2) Mixed macropore flow (low and high interaction): water flow in macropores, where a heterogeneous soil matrix leads to a mix of low and high lateral interactions between the macropore water and the soil matrix. The characteristic dye pattern shows a mix of broader and thinner individual stains.
(3) Macropore flow with high interaction/finger flow: This class includes water flow in macropores, where a permeable soil matrix leads to high lateral interactions between the macropore water and the soil matrix. But also smaller sized finger-shaped flow paths in water repellent soils or at texture transition zones, and finger flow caused by instabilities of the wetting front. This flow type class also includes funnel flow, which results in finger-shaped flow paths.
The characteristic dye pattern shows broader vertical elongated coherent stains.
(4) Heterogeneous matrix flow/ finger flow: water flow in a heterogeneous soil, water repellent soil, coarse textured soils, or soils with texture transitions. Characteristic dye pattern shows an infiltration front that proceeds with instabilities
(5) Homogeneous matrix flow: water flow in permeable soil. Characteristic dye pattern shows a homogeneous staining of the soil matrix

 Do these indices depend on the effective width of the profile? If there is a flow restriction, for example from a rock, wouldn't that lead to a "compression" of the water flux through the narrower width? Is it possible that water from one experiment gets drawn into another subplot through matric forces?

The indices are related to the actual width of the profile. The presence of stones or larger rock can lead to a redirection or funneling of the water flow. If the water flow is redirected into thinner flow paths, funnel flow will have the same flow pattern as finger flow. Since the focus in our study is on the flow type patterns and not specifically on the mechanisms causing the patterns, we combine both flow type classes "funnel" and "finger flow" into the class finger flow. Thus, the presence of rocks does not interfere with our flow type classification based on the image analysis. We will include a more detailed definition of the flow type classes in the revised manuscript, which we also explain in further detail in the comments to Reviewer 2.

In theory, it would be possible for water to be drawn from one subplot into the neighboring subplot. Due to the high vertical conductivities and the relatively high antecedent soil moisture with correspondingly low matric potentials, this probability is considered to be low. However, to avoid possible boundary effects, i.e. interactions between the subplots or between the plots and the outer (not irrigated) boundaries, we excluded a buffer of at least 6 cm width between neighboring subplots and at least 5 cm to the outer plot boundaries from the image analysis. We will include this information in the revised manuscript.

 Given the low n, there is a chance that the trend is random, even if it's statistically significant, no?

We actually consider n between ~4400 and ~6700 to be quite sufficient for the statistical test to be reliable. To test for significant differences in observed infiltration depths among age classes and among irrigation intensities we used the non-parametric Mood's median test.
The Mood's median test compares median pairs of two or more groups. A p-value lower than 0.05 indicates that at least the median of one group is significantly different from the other groups. One group of observations consists of the observed maximum staining depth at each pixel column (~400) per profile (5) and plot (3). Which leads to groups with a size between n=~4400 and ~6700.

Fig 4: When there were rocks at or immediately below the surface, what happened to the water that couldn't access that space? Did it run off?

We observed infiltration of water next to the rocks at the surface and along the stones below the surface. So the rocks lead only to very local (micro-topography-scale) redistribution but not to runoff. Significant surface runoff only occurred at the bare plots at the young moraines. However, we relate the higher surface runoff here to the process of structural sealing, since the vegetated plots unaffected by structural sealing had a similar amount of rocks, but lower surface runoff.

Page 12, Lines 16-18: Which appears to be the case for most profiles…?

Yes this is the case for most of the profiles, but mostly pronounced in 110a: plot 3, 4.9ka: plot 1 and 2, 13.5k: plot 1.  We consider to exclude the Pdye analysis in the revised manuscript as it adds little value (also noted by Reviewer 2).

There is still a reduction in SAD with depth, but SAD values in 50 and 80 cm at the older moraines tend to be higher than at the younger moraines.

Figure 6 and Page 14, Lines 6-7: Curious to read why sometimes SAD is greater for lower intensities and sometimes greater at higher intensities.

We discuss this issue on page 23-25:
We observed a decrease in SAD with intensity at all depths for the young moraines and an increase at the old moraines (50-60cm at 4.9ka and entire profile at 13.5ka).
We found, similarly to other studies investigating finger flow, that the number of fingers increases with higher flux rates.
At the rather homogeneous material of the young moraines we observed a decrease in SAD due to the merging of flow paths.

Page 18, flow type classification: A table with the percentages would be good here so that the reader doesn't have to piece together everything. Fig 10 is nice, but it is a little difficult to compare the length of the bar sections after "Matrix flow between rocks". Maybe something for an appendix.

We agree and will include a table with the frequency of the flow types in the revised manuscript.

Page 20, Fig 11: I'm wondering if the percentage of rocks in a profile affects this as well. This harkens back to my earlier comment in which way rocks affect all these flow characteristics.

The impact of rocks is an interesting point. The presence of rocks can divert water transport and create funnel flow, which we classify as finger flow (finger-shaped flow). The presence of larger stones can therefore lead to more preferential water transport. We would also have liked to have done a more in-depth analysis of the impact of stone content. However, a quantification of the real stone content in the soil for such an analysis is rather difficult due to the image analysis method used. The volume density profiles of the stones shown in Figure 4 in the manuscript are only subjective approximations, since the stones were defined manually during image analysis as described in the methods section. The difficulty lies in manually recording all the stones in the coarse moraine material that have a relevant influence on the flow paths.

Page 21, Lines 5-8: I was thinking about this the entire time while reading the manuscript. Can you estimate infiltrated volume or surface runoff? I would imagine the rocks close at the surface play a huge role here and not just the soil properties.

As mentioned above, stones at the surface usually only lead to very local, small scale redistribution and preferential infiltration instead of surface runoff. Surface runoff was only observed at the young moraines. Attempts to quantify this surface runoff unfortunately failed. Thus, we can only provide visual observations of the occurrence and relative amount of surface runoff. We also see the point that the stone content might impact the infiltration amounts especially at the youngest moraines with a high stone content at the surface.

However, in the manuscript we point out that we observed at both young moraines different amounts of runoff at different plots of the same age class, despite the plots having similar stone coverages. Stone coverage thus seems not to be the controlling factor here. Instead we relate the differences within the age classes to the vegetation cover and the soil properties.

Page 23, Lines 1-3: This makes sense to me. It could be a combination of both larger diameter and longer roots.

Indeed, the larger diameters are also a valid point, which we will include in the revised manuscript.

Page 23, Lines 10-16: I like that you bring this up. My initial interpretation would have been that the different external factors of the sites (which also affect landscape evolution) are more important than age.

This is unfortunately the drawback of the chronosequence approach in general. This space-for-time substitution approach assumes that for a sequence of sites (in this case moraines) with similar characteristics such as topography, climate, and parent material on which the soil was formed, time can be treated as the only variable. It is well known that the application of this chronosequence concept has some limitations as landscape development is much more complex than in form of a monotonic progression. The assumption that time is the only factor affecting soil development in a spatial sequence of soils is often the only option for a detailed historical tracking of landscape development at a particular location and thus still a fundamental tool for representing temporal changes in the Earth surface system (Phillips, 2015).

We will also include a statement regarding the assumptions of the chronosequence approach in the methods section of the revised manuscript.

Page 23, Lines 30-31: Didn't you argue on the previous page that hydrophobicity could affect the infiltration patterns…?

That is correct. Despite the fact the matric potential did not drop below field capacity we believe that the release of hydrophobic compounds during the decay of litter or by root activities create a hydrophobic soil matrix, which is due to variations in soil moisture from time to time more or less pronounced.

Page 25, Lines 29-30: This is an important point that needs to be included in the methods (unless I missed it somehow). Does this only refer to the outer edges or to the inner boundaries as well? How much was excluded?

This also accounts for the inner boundaries. In between neighboring subplots a buffer with a width of 6 cm was excluded from the analysis. We will include this information in the methods section in the revised manuscript.

Musso, A., Lamorski, K., Sławi´nski, C., Geitner, C., Hunt, A., Greinwald, K., and Egli, M., 2019: Evolution of soil pores and their characteristics in a siliceous and calcareous proglacial area, CATENA, 182, 104 154, https://doi.org/10.1016/j.catena.2019.104154.

---

## Author Comment (AC2)

**Response to Reviewer comments**

**Response to Reviewer 2 (John R. Nimmo)**

General Comments
This paper provides an extensive and valuable set of field observations of the subsurface flow patterns generated by three different irrigation intensities over four members of a soil chronosequence. As in previous works using similar methods, this study offers quantitative analysis of unsaturated flow features that otherwise would be evaluated subjectively and without quantification.

The main value is in providing evidence to elucidate how factors including soil age, input intensity, vegetative cover, and others influence the depth and homogeneity of the distribution of the infiltrated water. In particular, a major issue is the distinction between preferential and homogeneous flow patterns, understanding of which has tremendous importance to water supply and water quality matters, as well as to agriculture and ecosystem health. The linkage to physical phenomena is primarily through classification into six categories based on a modified version of the scheme of Weiler and Flühler (2004).
The paper provides useful documentation of soil developmental processes over 13500 years. Together with an earlier study of Hartmann et al. (2020a), it provides evidence of the differences resulting from calcareous-vs.-siliceous parent materials.

The authors would like to thank John R. Nimmo for spending his time to review and make valuable comments to improve our manuscript. We highly appreciate the detailed feedback and recommendations to improve the consistency of our manuscript.
We will address the comments and suggestions below.

The data analysis is extremely thorough. A great variety of statistical methods are employed, perhaps more than necessary. I see little or no value in the Pdye analysis because the constraint of monotonicity is a serious shortcoming that could distort the interpretation of how water behaves in the profile.

We will exclude the Pdye analysis in the revised manuscript, as we also see the point that it adds only little value.

Presentation of multifactor comparisons of many individual experiments is unavoidably complex, and is done here (figures 3-10) through an organization that requires the reader's time and effort to understand and evaluate, but it does show the results in a way that the effects of soil age, irrigation intensity, and spatial variability can be directly seen.

The main problem I find in the manuscript is confusion and inconsistency concerning the classification of types of subsurface flow. Much of this relates to the term "finger flow", for which I don't find a clear definition in this paper, and which seems to be used in different ways.

Some background from my own understanding: Three main categories of preferential flow are commonly used—funneled flow, which is directed into particular downward paths as a result of heterogeneities of the medium that provide faster flowpaths through the more conductive material; fingered flow, which is initiated at flow instabilities in the wetting front and sustained in downward preferential paths by the greatly enhanced hydraulic conductivity of the newly wetted material; and macropore flow, which proceeds through elongated continuous pores over significant distances within the medium.

We understand the problem with a missing definition of the used terms. We will include a more specific definition of the flow type classes in the revised manuscript and will change Page 7, Line 24+ to:

"[...] This classification method distinguishes between five flow types: (1) macropore flow with low interaction, (2) mixed macropore flow (low and high interaction), (3) macropore flow with high interaction, (4) heterogeneous matrix flow/finger flow, and (5) homogeneous matrix flow. We define macropore flow as water transport via root channel, earthworm burrows, and flow along fissures largely bypassing the matrix. The characteristic dye pattern shows narrow but long individual stains, which can be broader due to interactions with the surrounding soil matrix.

The term finger flow here summarizes all flow types that cause finger-shaped flow patterns, which includes finger flow caused by flow instabilities in the wetting front (Nimmo, 2021), finger-shaped flow paths due to water repellency, air entrapment or textural layering (Hendricks, 2001) and also funneled flow leading to vertical elongated finger-like flow paths. The latter is caused by the redirection and funneling of water by textural boundaries and large rocks (Hendricks, 2001) or by the heterogeneity of soil hydraulic properties (Nimmo, 2021). The characteristic flow patterns of all these flow types are very similar and thus cannot be distinguished by the image analysis: they show broader, vertically elongated, coherent flow paths, which indicate a preferential vertical water transport and leave large parts of the soil matrix dry. Dye patterns, which could not be classified as one of the five flow types were categorized as undefined.

We used a modified version (Hartmann et al., 2020a) of this classification which was more suitable for stony alpine soils. In the case of homogeneous matrix flow, the modified classification avoids that a high stone content leads to the detection of a heterogeneous flow pattern by breaking up the coherent stained area into smaller pieces, which then could be falsely classified as heterogeneous matrix flow, finger flow, or macropore flow depending on the abundance of rocks. In this case the flow type is assigned to a new flow type class called (6) 'homogeneous matrix flow between rocks'.

The modified classification also avoids a clear differentiation between 'macropore flow with high interaction' and 'finger flow'. As the original classification assigns finger-shaped flow paths only when both, the medium-sized stained path width (20–200 mm) and the biggest stained path width class (> 200 mm) account for approximately half of the dye coverage, fingers with smaller widths were not detected as such and automatically counted as macropore flow with high interaction. Hartmann et al., 2020a observed that finger-like flow paths with smaller widths were frequently present in alpine soils. Their dye patterns and distributions of stained path width classes are similar to 'macropore flow with high interaction'. Both classes cannot be distinguished from each other in the image analysis. Thus we renamed this class to (3) 'macropore flow with high interaction/ finger flow'. The classification was done for each pixel row per profile. [...]"

I see these categories to be represented in the scheme of Weiler and Flühler (2004) (hereafter referred to as WF2004), which is designed specifically for use in interpreting dye-tracer results. Macropore flow needs matrix interaction to be visible, as acknowledged in the first three categories of WF2004. I see the term "matrix heterogeneous flow" as a synonym for funneled flow, and it is quite adequate in that usage. Instability-initiated fingered flow would be difficult or impossible to distinguish from matrix heterogeneous flow when the only evidence is from pictures of dye-tracer distribution. Thus it is appropriate to group both of these flow modes together as in the fourth WF2004 category, "Heterogeneous matrix flow and fingering". Absence of preferential flow is reasonably called homogeneous matrix flow in the fifth category. In the present study, the use of the WF2004 classification scheme is a suitable approach for evaluating dye-tracer patterns in terms of preferential flow. It is extended reasonably with the added sixth category to accommodate effects of large stones in the soil.

The other modifications adopted here are poorly explained, and appear to deviate significantly from some widely understood general features of preferential flow, and from the evidence available from this study as I understand it. Below, I explain these issues further in relation to finger flow and macropore flow.

Finger flow
Instability-initiated fingers are possible, though my expectation in such heterogeneous soil is that these are likely to be rapidly channeled into funneled flowpaths. Based on the images and other available information in the present study, I doubt that it is possible to discern whether instability-initiated fingering is an active process. In 18:24 (location noted as page:line) the term "finger flow" seems to mean any preferential flow that is identified by finger-like patterns of dye tracer, not limited to the downward-moving fingers of wetness generated at a wetting-front instability. The finger-like patterns in the dye could result from other modes of preferential flow. If what is meant is just that the patterns have a finger-like shape, without regard to specific process, "finger flow" would be better replaced by the general term "preferential flow". This issue occurs also in 1:14, 21:4, 22:3-6, 22:16, 23:21-22, 24:5-8, and 26:4-13.

We agree that is was not made clear in the manuscript that we are talking about the shape of the flow pattern when referring to finger flow and not specifically to finger flow in the narrow sense (generated at a wetting-front instability). We will address this issue by explaining our definition of the flow classes as written above and will further include the following changes into the revised manuscript:

Page 1, Line 14: we will change "finger flow paths" to "finger-shaped flow paths"
The same changes will be done on the following pages and lines: 21:4, 22:3-6, 22:16, 23:21-22, 24:5-8, and 26:4-13

On the other hand, the specific mode of instability-initiated finger flow is the subject of 23:4-6 and 24:11—25:2. It also is strongly related to the effects of hydrophobicity in 22:6 – 23:9. These passages need clarification and consistency. Overall, finger flow must be explicitly defined and the term used consistently. If the paper actually does claim that instability-generated finger flow is detected in these experiments, there needs to be justification for how this can be determined.

We see the point that using the term finger flow for finger-shaped flow paths without a previous definition what we count as finger flow leads to misunderstandings and inconsistencies.
We will replace the term "finger flow" in the listed sections with the term "finger-shaped flow paths" to make it clear that we are talking not only specifically about finger flow in the narrow sense, but name all possible reason that could have caused the development of finger-shaped flow paths.

Macropore flow
There needs to be more discussion of the possible effects of macropores. The soils are likely rich in narrow macropores that result from growing and decaying roots (apparent in the images of both young and old soils), and other bioactive processes. If such macropores convey significant water that then has some degree of interaction with soil matrix material, they could create flow pattern features of the types observed. The statements in 18:15-22 are hard to understand and accept, where it is implied that finger flow can be distinguished from macropore flow, and stated that no macropore flow was found. If there are reasons to justify ruling out active macropore flow, they need to be carefully explained.

We see the point that we cannot rule out macropore flow in general, since macropores due to bioactivity (e.g. roots) are present and also very likely to conduct water. However, after carefully screening the photos, and from our observations during the excavation we cannot determine conclusively whether the staining pattern is a result of macropore flow with high interactions or whether finger-shaped flow paths, caused by a variety of site conditions, superimpose the water transport in existing macropores. Thus, we will change the statement about the possibility of an impact of macropore flow in the revised manuscript:

Page 18, Line 14+

"[..] At the youngest moraine matrix flow is the predominant flow type (relative frequency > 0.6) followed by the flow type class 'Macropore flow with high interaction/ Finger flow'.
A reliable distinction between macropore flow with high interaction and finger-shaped flow could neither be made through the image analysis nor through on-site assessment. As narrow macropores were present (e.g. thin root channels), they certainly also contribute to water transport, but it is also likely that this process is overlaid by finger-shaped flow paths, caused by site conditions. Since the water transport patterns of both flow types cannot be distinguished and show finger-shaped flow patterns, they are also referred to as finger-shaped flow in the following.
At the 160a the relative frequency of matrix flow decreased to 0.5 and the frequency of finger-shaped flow increased. At the two oldest moraines the dominant flow type is flow with finger-shaped flow paths and the relative frequency of matrix flow dropped below 0.3. [..]"

I cannot make sense of the statements in 7:30-33, which seem to imply that finger flow can be distinguished from macropore flow, but then contradict that in saying that no such differentiation is made. Then there is confusion in the statement that narrow finger flowpaths could somehow be misclassified as macropore flow with high (but not low or intermediate) interaction.

We will clarify the origin of this joint flow type class, as described above. We further will weaken the statement that macropore flow could be ruled out. We further include the statement that both flow class types cannot be distinguished from each other in the image analysis.

Page 7, Lines 30+ new phrasing:
"[..] The modified classification also avoids a clear differentiation between 'macropore flow with high interaction' and 'finger flow'.  As the original classification assigns finger-shaped flow paths only when both the medium-sized stained path width (20–200 mm) and the biggest stained path width class (> 200 mm) account for approximately half of the dye coverage, fingers with smaller widths were not detected as such and automatically counted as macropore flow with high interaction. Hartmann et al., 2020a observed that finger-like flow paths with smaller widths were frequently present in alpine soils. Their dye patterns and distributions of stained path width classes are similar to 'macropore flow with high

interaction'. Both classes cannot be distinguished from each other in the image analysis. Thus we renamed this class to (3) 'macropore flow with high interaction/ finger flow' […]".

Section 3.2 (18:12-30) needs to be rewritten for consistency with other clarifications. The category "Macropore flow with high interaction/ Finger flow" is mentioned here and in Figure 10, but it is not mentioned in the definition of the categories on page 7 and is not in the scheme of WF2004.

The class "Macropore flow with high interaction/ Finger flow" is part of an adapted version of the scheme of WF2004 by Hartmann et al., 2020 to also include smaller sized finger-shaped flow paths. We will include this clarification into the revised manuscript as stated above.
In the revised manuscript we will rewrite section 3.2 (18:12-30):

"[…] Using the VD-profiles of the three SPW classes and their proportion on the total dye coverage to characterize flow types (Weiler, 2001) we found that over the millennia flow types transition from matrix flow to preferential flow in form of finger shape flow paths (Figure 10 a).
At the youngest moraine matrix flow is the predominant flow type (relative frequency > 0.6) followed by the flow type class 'Macropore flow with high interaction/ Finger flow'.
A reliable distinction between macrpore flow with high interaction and finger-shaped flow could neither be made through the image analysis nor through on-site assessment. As narrow macropores were present (e.g. thin root channels), they certainly also contributed to water transport, but it is also likely that this process is overlaid by finger-shaped flow paths, caused by site conditions. Since the water transport patterns of both flow types cannot be distinguished and show finger-shaped flow patterns, they are also referred to as finger-shaped flow in the following.
At the 160a the relative frequency of matrix flow decreased to 0.5 and the frequency of finger-shaped flow increased. At the two oldest moraines the dominant flow type is flow with finger-shaped flow paths and the relative frequency of matrix flow dropped below 0.3.
Considering the entire profile depth of 1 m, the frequency of matrix flow decreases and the frequency of finger shape flow paths increase continuously with moraine age. A depth differentiated view shows a higher proportion of finger-shaped flow at the 4.9ka than at the 13.5ka in the upper 20 cm (Figure 10 a1). In the other depths (Figure 10 a2 to a4), however, a continuous increase in finger-shaped flow frequency with moraine age was observed. With regard to the irrigation intensity no consistent impact on the flow type distribution across the millennia could be identified (Figure 10 b). At the 110a and 160a moraine the two dominant flow types (matrix flow and finger-shaped flow paths) show an almost equal distribution across all irrigation intensities. A tendency to less matrix flow is observed at the 4.9ka, whereas at the 13.5ka the frequency of matrix flow increases with increasing irrigation intensity. Differentiated by depth, we observed no systematic trend in flow type frequency distribution with increasing irrigation intensity in the upper 20 cm for all age groups (Figure 10 b1). From a depth of 20 cm, the 4.9ka and 13.5ka each show a trend-like behavior in the shift of the frequency distribution with irrigation intensity analogous to the observation of the entire soil profile (Figure 10 b2 to b4). From a depth of 40 cm, the relative frequency of matrix flow also increases with increasing irrigation intensity at the 110a and 160a (Figure 10 b3 to b4). […]"

Overall:
This paper is dense with useful information and provides insights into the development of preferential flow paths during landscape evolution and several other important facets of unsaturated flow in calcareous soils. It needs revision for consistency and adherence to evidence and general understanding of the different types of preferential flow paths. Because the basic experimental work and presentation of data are sound, I have classed these revisions as minor, though I see them as extremely important.

Specific Comments

6:16-18. Rewrite for clarity. Use of "below" in line 16 suggests that the excavation is downward to produce horizontal planes, but "vertical profiles" in 17 suggests otherwise. Does "below" mean "downslope of"? The operation suggests that a trench was first excavated off to the side of the plot to provide access for vertical profiling. More details on this would be helpful.

We will clarify in the revised manuscript:
"A first vertical profile was excavated 10-15 cm downslope of the lower edge of the irrigated plot to check for subsurface lateral flow."

7:17. What is meant by "amount"? The number of flow paths?

We will clarify in the revised manuscript:
"The surface area density (SAD) is an indicator for the number of individual flow paths and was calculated for each pixel row of the five profiles by using the intercept density, which describes the number of interfaces between stained and unstained pixels divided by the horizontal width of the soil profile."

7:28-29. Clarify—maybe make two sentences. Start with a clear description of the problem caused by rocks. Then the solution devised.

We will clarify in the revised manuscript as also described above:
"We used a modified version (Hartmann et al., 2020a) of this classification which was more suitable for stony alpine soils. In case of homogeneous matrix flow, the modified classification avoids that a high stone content leads to the detection of a heterogeneous flow pattern by breaking up the coherent stained area into smaller pieces, which then could be falsely classified as heterogeneous matrix flow, finger flow, or macropore flow depending on the abundance of rocks. In this case the flow type is assigned to a new flow type class called (6) 'homogeneous matrix flow between rocks'."

7:31-33 Why "misclassified"? What is unreasonable about "macropore flow with high interaction"?

As stated above, we will remove this statement and clarify in the revised manuscript as follows:

Page 7, Lines 30+:
"The modified classification also avoids a clear differentiation between 'macropore flow with high interaction' and 'finger flow'. As the original classification assigns finger-shaped flow paths only when both the medium-sized stained path width (20–200 mm) and the biggest stained path width class (> 200 mm) account for approximately half of the dye coverage, fingers with smaller widths were not detected as such and automatically counted as macropore flow with high interaction. Hartmann et al., 2020a observed that finger flow paths with smaller widths were frequently present in alpine soils. Their dye patterns and distributions of stained path width classes are similar to 'macropore flow with high interaction'. Both classes cannot be distinguished from each other in the image analysis. Thus we renamed this class to (3) 'macropore flow with high interaction/ finger flow'."

7:34. Proportion in relation to what? PFF needs to be defined more clearly.

We will clarify in the revised manuscript:

"A preferential flow frequency index (PFF) was calculated based on the frequency of preferential flow type classes (1-4) across the profile at each experimental plot. As the flow type classification was done for each pixel row, the PFF is the number of pixel rows classified as a preferential flow type (1-4) divided by the total number of pixel rows."

12:8-10. Split sentence into two, for clarity.

We will clarify in the revised manuscript:

"At the 4.9ka and 13.5ka the fraction of SPW>200 mm is lower in the upper 10-20 mm compared to the young moraines. Over the entire profile range, path widths of the category 20<SPW<200 mm most often have the largest share on the dye coverage."

14:1. Replace "Whereas" with "In contrast," or similar expression.

We will update the sentence in the revised manuscript to:

"In contrast the old moraines have a high fraction of 20<SPW<200mm combined with a high SAD, which indicates a higher number of smaller, narrow blue-colored areas and thus more individual active flow paths at the older moraines and less individual flow paths but larger continuous areas used for water transport at the young moraines."

17:4-5. It seems at best to be a very subtle effect for the middle portion of the profiles to be less significantly different. Maybe not worth mentioning.

We will exclude this statement in the revised manuscript.

We will update the sentence in the revised manuscript to:

"Only larger stones and occasional clay lenses (the size of a few centimeters) or other material heterogeneities create heterogeneous matrix flow."

We will update the sentence in the revised manuscript to:

"At both moraines deep infiltration, almost no surface runoff, and no subsurface lateral flow was observed."

22:4-5. The point is not about the water transport in general but the pattern of the water transport that is affected. Insert "pattern" or some similar expression.

We will update the sentence in the revised manuscript to:

"Thus, water infiltrated heterogeneously and/or the water transport pattern was affected by properties of the soil surface or of the upper soil layer."

We will update the sentence in the revised manuscript to:

"After more than 10000 years of landscape development, subsurface hydrology at the calcareous geology is ruled by finger-shaped flow and deep infiltration, whereas at the siliceous geology storage capacity in the top soil strongly increased with a corresponding reduction in infiltration depths and a shift to macropore flow."

We will correct "Matrix potential" to "Matric potential" in the revised manuscript.

23:32. The paragraph starting here, and also the next one, are all about the two older soils. This should be made clear to the reader in the first sentence at line 32. Consider rearranging discussion from this point through 25:13 in order to proceed in the logical order of young to old.

We will rearrange the discussion from page 23, line 32 to page 25, line 13 in the logical order of young to old:

"At the young moraines (110a and 160a), we observed an in increase in the frequency of matrix flow at greater depths with increasing irrigation intensity, which is caused by an increase in the SPW>200 mm with a simultaneous decrease in the median dye coverage and a decrease in the number of flow paths (SAD) (Figure 5e). The decrease in median dye coverage with increasing intensity is particularly pronounced at the bare plots (data not shown). No clear trend can be seen at the plots with a higher vegetation cover. However, the decrease in SAD and the increase in stained path widths indicates that water flow paths that reach greater depths tend to widen and to merge together with increasing irrigation intensity. This process might be facilitated by higher water contents at greater depths or by a change in material properties.

At the 4.9ka moraine, we also observed an increase in dye coverage (Figure 5 (a)) and an increase in the number of flow paths (Figure 5 (e)). The proportion of 20<SPW<200 mm increases (Figure 5 (c)) and the proportion of SPW>200 mm decreases (Figure 5 (d)), which then leads to an increase in the frequency of finger flow paths in the flow type classification (Figure 10 (b2) to (b4)).

At the 13.5ka we observed an increase in dye coverage (Figure 5 (a)), an increase in infiltration depth (Figure 3), a broadening of the flow paths (Figure 5 (c-d)), and an increase in the number of flow paths (Figure 5 (e) and Figure 6) with increasing irrigation intensity. Other than at the 4.9ka, the increase in the proportion of SPW>200 mm on the dye coverage leads to a transition to more matrix flow (Figure 10) in the flow type classification.

However, the process of flow type classification is only based on the proportions of the three SPW classes on the dye coverage (Weiler, 2001). The number of flow paths or the dye coverage itself are not taken into account. We observed at both age classes that with increasing irrigation intensity more fingers are generated and more soil space is used for water transport (Figure 5). It also has to be stated that at sites where preferential flow occurs in form of macropore flow the relations can be different, since the controls inducing macropore flow are different (Nimmo, 2021). The formation of finger flow paths and their properties such as number, flow velocity, or width are in a complex interplay with the surrounding soil moisture, the flux rate and soil properties (Nimmo, 2021).
Studies focusing on the formation of finger-shaped flow paths found the finger width not only to be influenced by soil properties, and initial and boundary conditions (Glass et al., 1989), but also by the flow rate through the finger (Parlange and Hill (1976), White et al. (1976)) with higher flow rates leading to an increase in finger width. This was also observed by Ma et al. (2008), who also found a positive correlation between rainfall intensity, time of finger flow occurrence and mean velocity. The increase in mean velocity of the fingers leads to a faster downward transport and thus deeper infiltration depths with higher irrigation intensities (Cremer et al., 2017). An increase in the number of fingers with higher flux rates was also observed (Sililo and Tellam, 2000). These findings by other studies are similar to our observations at the 13.5ka moraine. It is unclear what causes the different observations in the dominant flow path widths at the 4.9ka and 13.5ka moraines. We can only speculate whether the higher organic matter content, the higher root density, or soil properties such as the lower hydraulic conductivities and

higher porosity play a role in producing narrower flow paths with increasing irrigation intensity at the 4.9ka moraine."

25:24-27. Confusing. Which of the plots were less affected by the direct application of water? Why is there consideration of the boundaries in this?

We mention the observations of the deep infiltration at the plot boundaries of the bare plots at both young moraines to support the assumption that water would have infiltrated deep into the soil, when the soil surface was not affected by structural sealing due to the irrigation process.
We will rephrase this part in the revised manuscript for clarification:

"We further assume that the irrigation with the hand-operated sprayer, which had to be held close to the soil surface due to strong winds, in part led to a high force of application and promoted structural sealing at the barer plots of the 110a and 160a moraines. At both moraines, deep infiltration was often found at the boundaries of the bare plots. Since the plot boundaries were not irrigated, they were also not affected by structural sealing. Water running off to the sides infiltrated deep into the soil. This observation suggests that a more homogeneous and deep transport of the water can take place in this quite homogeneous and unsorted material (Hartmann et al., 2020b), if the surface is not influenced by particle displacement.
Thus, it is assumed that the proportion of preferential flow paths at the young moraines is generally overestimated and homogeneous to heterogeneous matrix flow with deep infiltration are probably the dominant flow types under natural rainfall conditions. As the plot boundaries (outer boundaries and boundaries of neighboring subplots) are excluded from the image analysis to avoid edge effects, the here observed deep percolation could not be accounted for in our quantitative analysis."

---

## Author Response (AR1)

**Response to Reviewer comments**

**Response to Reviewer 1**

General Comments
Hartmann et al. present work on infiltration experiments across a moraine chronosequence in the Swiss alps, spanning almost 14,000 years. They performed infiltration experiments on four plots of different ages, with each plot being subdivided into three subplots where different precipitation intensities were applied.

The work is a heavily revised version of a previously submitted manuscript. I was one of the original reviewers back then and suggested a rejection. This submission was deemed different enough to be considered a new submission, and I would agree with this assessment. The way the authors reworked the current manuscript makes it much more enticing and sets it apart further from Hartmann 2020a and 2020b (in my opinion).

The authors would like to thank the reviewer for spending her/his time on reviewing our revised manuscript. We are pleased that the changes to the original manuscript were well received and appreciate the feedback and interesting comments.

In the introduction, the authors describe the need for identifying different flow patterns for potential integration into landscape evolution models. It would be interesting to revisit this idea in the discussion. What does the work suggest such an integration could look like, and more importantly, what would the practical differences be between the different flow types for such a model (especially considering that some of the differences between the plots appear minor, even though they were found to be significant)?

We found that, due to the complexity of landscape development, time should not be regarded as the only primary evolutionary factor. Thus, it would be necessary/helpful to intensify correlation studies, such as those already vaguely tackled in Figure 11, in order to integrate a corresponding modelling approach in SLEMs. The aim of this study would be to identify the main influencing factors (e.g. soil properties, vegetation characteristics) on the formation of preferential flow paths and to create transfer functions between properties and the degree of preferentiality. The derived preferential flow frequency index (PFF) could be a helpful variable to quantify the extent of preferential water transport. The minor differences in the profiles between the plots would therefore not be of great importance.
How the final exact implementation of the description of the flow types in the model could look like largely depends on the desired complexity/accuracy of the target model (empirical/physically based).

Since this is an interesting point, we included this as an outlook into the conclusion on page 27, line 31 to page 28, line 8:
"Our study made it clear that, due to the complexity of landscape development, time should not be regarded as the only primary evolutionary factor. The geology and the resulting landscape properties (e.g. soil structure and texture, vegetation, organic matter content, etc.) have a major impact on the development of subsurface hydrological flow paths. The type of hydrological flow paths significantly influences the redistribution of solid and solutes and thus also affects landscape development. The feedback between soil hydraulic process and soil structure is therefore an important aspect that needs to be included in SLEMs. Therefore it is necessary to intensify studies on the main influencing factors

(e.g. soil properties, vegetation characteristics) on preferential flow path formation. Transfer functions between properties and the degree of preferentiality could be one possible approach to integrate such a modelling approach into SLEMs. Our derived preferential flow frequency index (PFF) could be a helpful variable to quantify the extent of preferential water transport. The here provided data and observations can thus help to improve the handling of hydrologic processes and their role within the feedback cycle of the hydro-pedo-geomorphological system when it comes to soil and landscape evolution modeling."

I am also wondering if the amount of rocks has an effect on the flow type. If a large fraction of the profile is taken up by rocks, percolating water will be restricted to the space between the rocks. The authors do include flow types that take into account rocks in the profile. From what I understand, though, this applies mostly to homogeneous flow that happens around the rocks. What would happen if a larger rock led to an effective partitioning of an otherwise homogeneous wetting front? (That is, if the soil below the rock remained dry)

In the case of large rocks leading to partitioning / funneling of the wetting front, this funnel flow is automatically counted as finger-shaped flow within the image analysis (when the redirection of water leads to thinner flow paths). We clarified in the revised manuscript on page 8, line 15 to 21 that the flow type class finger flow refers to the water transport pattern and not specifically to the narrow sense of fingers created only by instabilities of the wetting front, and thus also includes funnel flow due to rocks. However, within the scope of this study, it is not possible to differentiate the causes of the observed finger-shaped flow paths, although the influence of stone frequency on the occurrence of finger-shaped flow paths is an interesting point. The overall effect, that water is transported preferentially via finger shaped flow paths to greater depths, remains the same regardless of the cause.

Further, it appears that some profiles exhibited a significant portion of rocks in the upper soil layers. Overland flow was not measured, but it could be beneficial to talk a little more about the potential impact of less water infiltrating at these sites.

A possible impact of stone/rocks at the surface is a very important aspect. We also see the point that the stone content could impact the infiltration amounts, especially at the youngest moraines, which are characterized by a high stone content at the surface. However, our observations did not show a direct relation between the stone content and the reduced infiltration that occurred at some plots at the young moraines, as the stone content was roughly similar at all three plots per age class.
We included a statement regarding the impact of stones in the revised manuscript on page 22, line 12 to 16:

"The surface of the young moraines is characterized by high stone deposits (Figure 12), but since the stone cover was roughly similar at all three plots per age class we do not assume that the differences in surface runoff are related to the amount of stones at the surface. We observed that stones at the surface mostly result in only very local, small scale redistribution and preferential infiltration instead of surface runoff. We therefore link the differences in infiltration patterns and resulting staining patterns to the differences in vegetation cover. […]"

In the discussion, the authors mention very briefly that the edges of the plots were not analyzed. I might have missed this earlier, but does this only apply to the outer edges of the 1x1.5 m plots or also to the borders between plots 1 and 2 and plots 2 and 3? If so, how big of a buffer was included? I could imagine that interactions around the inner boundaries could have an impact, too.

To avoid a possible impact by interactions around the inner boundaries we excluded a soil space between neighboring subplots with a width of at least 6 cm from the analysis. We included this information in the method section in the revised manuscript on page 7, line 17-19:

"To avoid possible interference from interactions around the inner and outer subplot boundaries, a buffer with a width of 6 cm between adjacent subplots and 5 cm to the outer plot boundary was excluded from the analysis."

I think the revisions are a little more than just minor, but I am confident that the authors can address them.

Specific Comments
Page 4, Lines 9-11: What is the reasoning for having two plots of ~the same age?

The original reason for selecting the 110 year old moraine as the youngest and the 160 year old as the second youngest moraine was mainly the result of local conditions at the glacier forefield. The actual goal was to select age groups that were as identical as possible to the moraine ages selected in Hartmann et al., 2020a (30a, 160a, 3000a, 10000a). This was not entirely possible for the youngest moraine. We added an explanation in the revised manuscript on page 4, line 18 to 20:

"The choice of the 110a moraine as the youngest moraine is the result of the local conditions, as no adequate moraine younger than that could be identified that also ensured comparability in terms of elevation and microclimate (Musso et al., 2019).
We therefore had to compromise and selected the moraine with an age of 110 years as our youngest moraine."

Page 7, Lines 24+ Can you describe what the practical differences are between the flow types?

We have expanded our explanation of the flow types on page 8, line 3-21 to now to include the following:

"This classification method distinguishes between five flow types: (1) macropore flow with low interaction, (2) mixed macropore flow (low and high interaction), (3) macropore flow with high interaction, (4) heterogeneous matrix flow/finger flow, and (5) homogeneous matrix flow. The five flow types differ in the spatial extent of the water transport and thus in the proportion of the involved soil matrix.  This has an impact on flow velocities, water availability and solute transport.  From (1) to (5) the preferentiality of the water transport decreases and the homogeneity and spatial water availability increases.
We define macropore flow as water transport via root channels, earthworm burrows, and flow along fissures largely bypassing the matrix. The characteristic dye patterns show narrow but long individual stains, which in some cases can be broader due to interactions with the surrounding soil matrix. The level of lateral interactions between the water transported via macropores and the surrounding soil matrix mainly depends on the soil matrix. In low permeable or saturated soils the lateral interactions are usually low and the characteristic dye patterns show narrow but long individual stains. Permeable soils enable

high lateral interactions between macropore water and the soil matrix. The dye patterns show broader individual stains.

The term finger flow here summarizes all flow types that cause finger-shaped flow patterns, which includes finger flow caused by flow instabilities in the wetting front (Nimmo, 2021), finger-shaped flow paths due to water repellency, air entrapment or textural layering (Hendrickx and Flury, 2001) and also funneled flow leading to vertical elongated finger-like flow paths. The latter is caused by the redirection and funneling of water by textural boundaries and large rocks (Hendrickx and Flury, 2001) or by the heterogeneity of soil hydraulic properties (Nimmo, 2021). The characteristic flow patterns of all these flow types are very similar and thus cannot be distinguished by the image analysis: they show broader, vertically elongated, coherent flow paths, which indicate a preferential vertical water transport and leave large parts of the soil matrix dry."

Page 7, Lines 31 Do these indices depend on the effective width of the profile? If there is a flow restriction, for example from a rock, wouldn't that lead to a "compression" of the water flux through the narrower width? Is it possible that water from one experiment gets drawn into another subplot through matric forces?

The presence of stones or larger rocks can lead to a redirection or funneling of the water flow. If the water flow is redirected into thinner flow paths, funnel flow will have the same flow pattern as finger flow. Since the focus in our study is on the flow type patterns and not specifically on the mechanisms causing the patterns, we combine both flow type classes "funnel" and "finger flow" into the class "finger-shaped flow", as the overall effect of preferential flow to greater depth remains the same.
The flow type classification is based on the width of individual flow paths, thus neither the actual width of the profile nor the width of the profile reduced by the stones (effective width) have an impact on the flow type classification.

In theory, it would be possible for water to be drawn from one subplot into the neighboring subplot. Due to the high vertical conductivities and the relatively high antecedent soil moisture with correspondingly low matric potentials, this probability is considered to be low. However, to avoid possible boundary effects, i.e. interactions between the subplots or between the plots and the outer (not irrigated) boundaries, we excluded a buffer of at least 6 cm width between neighboring subplots and at least 5 cm to the outer plot boundaries from the image analysis. We included this information in the method section in the revised manuscript on page 7, line 17-19:

"To avoid possible interference from interactions around the inner and outer subplot boundaries, a buffer with a width of 6 cm between adjacent subplots and 5 cm to the outer plot boundary was excluded from the analysis."

Page 9, Lines 13 Given the low n, there is a chance that the trend is random, even if it's statistically significant, no?

We consider a sample size n between ~4400 and ~6700 to be quite sufficient for the statistical test to be reliable. To test for significant differences in observed infiltration depths among age classes and among irrigation intensities we used the non-parametric Mood's median test.
The Mood's median test compares median pairs of two or more groups. A p-value lower than 0.05 indicates that at least the median of one group is significantly different from the other groups. One

group of observations consists of the observed maximum staining depth at each pixel column (~400) per profile (5) and plot (3). Which leads to groups with a size between n=~4400 and ~6700.

Fig 4: When there were rocks at or immediately below the surface, what happened to the water that couldn't access that space? Did it run off?

We observed infiltration of water just next to the stones at the surface and along the stones/rocks below the surface.  So the stones lead only to very local (micro-topography-scale) redistribution but not to runoff. Significant surface runoff only occurred at the bare plots at the young moraines. However, we relate the higher surface runoff here to the process of structural sealing, since the vegetated plots unaffected by structural sealing had a similar amount of stones, but lower surface runoff.

We included a statement regarding the impact of stones in the revised manuscript on page 22, line 12 to 16:
""The surface of the young moraines is characterized by high stone deposits (Figure 12), but since the stone cover was roughly similar at all three plots per age class we do not assume that the differences in surface runoff are related to the amount of stones at the surface. We observed that stones at the surface mostly result in only very local, small scale redistribution and preferential infiltration instead of surface runoff. We therefore link the differences in infiltration patterns and resulting staining patterns to the differences in vegetation cover."

Page 12, Lines 16-18: Which appears to be the case for most profiles…?

Yes this is the case for most of the profiles, but most pronounced in 110a: plot 3, 4.9ka: plot 1 and 2, 13.5k: plot 1.  However, we excluded the Pdye analysis in the revised manuscript as it adds little value (also noted by Reviewer 2).

We see a reduction in SAD with depth at all age classes, but SAD values in 80 cm at the older moraines tend to be higher than at the younger moraines. We rephrased the sentence on page 14, line 6-7 in the revised manuscript to:

"At the young moraines, the SAD in the upper 10-20 cm is comparable to that of the old moraines, but decreases more strongly with depth (Figure 6).  In 80 cm, SAD seems to be higher at the older moraines."

Figure 6 and Page 14, Lines 6-7: Curious to read why sometimes SAD is greater for lower intensities and sometimes greater at higher intensities.

We discuss this issue in section 4.2 on page 24-26:
We observed a decrease in SAD with intensity at all depths for the young moraines and an increase at the old moraines (50-60cm at 4.9ka and entire profile at 13.5ka).
We found, similarly to other studies investigating finger flow that the number of fingers increases with higher flux rates.

At the rather homogeneous material of the young moraines we observed a decrease in SAD likely due to the merging of flow paths.

Page 18, flow type classification: A table with the percentages would be good here so that the reader doesn't have to piece together everything. Fig 10 is nice, but it is a little difficult to compare the length of the bar sections after "Matrix flow between rocks". Maybe something for an appendix.

We agree and included two tables giving the relative frequencies for the entire profile depth (Table A1) and for the four depth sections (Table A2) in the appendix on page 29 and 30.

Page 20, Fig 11: I'm wondering if the percentage of rocks in a profile affects this as well. This harkens back to my earlier comment in which way rocks affect all these flow characteristics.

The impact of rocks/stones in general is an interesting point. The presence of rocks/stones can divert water transport and create funnel flow, which we classify as finger-shaped flow. The presence of larger stones can therefore lead to more preferential water transport. We would also have liked to have done a more in-depth analysis of the impact of the stone content. However, a quantification of the real stone content in the soil for such an analysis is rather difficult due to the image analysis method used. The volume density profiles of the stones shown in Figure 4 in the manuscript are only subjective approximations, since the stones were defined manually during image analysis as described in the methods section. The difficulty lies in manually recording all the stones in the coarse moraine material that have a relevant influence on the flow paths.

Page 21, Lines 5-8: I was thinking about this the entire time while reading the manuscript. Can you estimate infiltrated volume or surface runoff? I would imagine the rocks close at the surface play a huge role here and not just the soil properties.

We also see the point that the stone content might impact the infiltration amounts especially at the youngest moraines with a high stone content at the surface. However, in the manuscript we point out that we observed at both young moraines different amounts of runoff at different plots of the same age class, despite the plots having similar stone coverages. Stone coverage thus seems not to be the controlling factor here. Instead we relate the differences within the age classes to the vegetation cover and the soil properties. Unfortunately, attempts to quantify this surface runoff failed. Thus, we can only provide visual observations of the occurrence and relative amount of surface runoff. But since the possible impact of stone/rocks at the surface is a very important aspect, we addressed it in the revised manuscript on page 22, line 12 to 16:

"The surface of the young moraines is characterized by high stone deposits (Figure 12), but since the stone cover was roughly similar at all three plots per age class we do not assume that the differences in surface runoff are related to the amount of stones at the surface. We observed that stones at the surface mostly result in only very local, small scale redistribution and preferential infiltration instead of surface runoff. We therefore link the differences in infiltration patterns and resulting staining patterns to the differences in vegetation cover."

Page 23, Lines 1-3: This makes sense to me. It could be a combination of both larger diameter and longer roots.

Indeed, the larger diameters are also a valid point, which we included in the revised manuscript on page 24, line 7.

Page 23, Lines 10-16: I like that you bring this up. My initial interpretation would have been that the different external factors of the sites (which also affect landscape evolution) are more important than age.

This is unfortunately the drawback of the chronosequence approach in general. This space-for-time substitution approach assumes that for a sequence of sites (in this case moraines) with similar characteristics such as topography, climate, and parent material on which the soil was formed, time can be treated as the only variable. It is well known that the application of this chronosequence concept has some limitations as landscape development is much more complex than in form of a monotonic progression. The assumption that time is the only factor affecting soil development in a spatial sequence of soils is often the only option for a detailed historical tracking of landscape development at a particular location and thus still a fundamental tool for representing temporal changes in the Earth's surface system.

We included a statement regarding the assumptions of the chronosequence approach in the revised manuscript on page 3, line 33:

"The chronosequence approach assumes that for a sequence of sites (in this case moraines) with similar characteristics such as topography, climate, and parent material on which the soil was formed, time can be treated as the only variable. It is well known that the application of this chronosequence concept has some limitations as landscape development is much more complex (Wojcik et al., 2021). The assumption that time is the only factor affecting soil development in a spatial sequence of soils is often the only option for the historical tracking of landscape development at a particular location and thus still a fundamental tool for representing temporal changes in the Earth's surface system (Phillips, 2015)."

Page 23, Lines 30-31: Didn't you argue on the previous page that hydrophobicity could affect the infiltration patterns…?

That is correct. Despite the fact the matric potential did not drop below field capacity we believe that the release of hydrophobic compounds during the decay of litter or by root activities create a hydrophobic soil matrix, where hydrophobicity shows an effect during drier conditions.

Our statement on the matric potential referred to measurements at depths of 10 to 50 cm. We observed that the matric potential at the three locations never dropped below field capacity and thus the soil moisture status between the locations in general did not show any significant differences. However, the observations do not rule out the possibility that, under the given weather conditions with high solar radiation and soil evaporation/transpiration, the soil moisture at the soil surface decreases, which could increase a possible hydrophobic behavior of the soil surface due to the high organic matter content.

Page 25, Lines 29-30: This is an important point that needs to be included in the methods (unless I missed it somehow). Does this only refer to the outer edges or to the inner boundaries as well? How much was excluded?

This also accounts for the inner boundaries. In between neighboring subplots a buffer with a width of 6 cm was excluded from the analysis. We included this information in the methods section in the revised manuscript on page 7, line 17-19:

"To avoid possible interference from interactions around the inner and outer subplot boundaries, a buffer with a width of 6 cm between adjacent subplots and 5 cm to the outer plot boundary was excluded from the analysis."

Musso, A., Lamorski, K., Sławi´nski, C., Geitner, C., Hunt, A., Greinwald, K., and Egli, M., 2019: Evolution of soil pores and their characteristics in a siliceous and calcareous proglacial area, CATENA, 182, 104 154, https://doi.org/10.1016/j.catena.2019.104154.

**Response to Reviewer comments**

**Response to Reviewer 2 (John R. Nimmo)**

General Comments
This paper provides an extensive and valuable set of field observations of the subsurface flow patterns generated by three different irrigation intensities over four members of a soil chronosequence. As in previous works using similar methods, this study offers quantitative analysis of unsaturated flow features that otherwise would be evaluated subjectively and without quantification.

The main value is in providing evidence to elucidate how factors including soil age, input intensity, vegetative cover, and others influence the depth and homogeneity of the distribution of the infiltrated water. In particular, a major issue is the distinction between preferential and homogeneous flow patterns, understanding of which has tremendous importance to water supply and water quality matters, as well as to agriculture and ecosystem health. The linkage to physical phenomena is primarily through classification into six categories based on a modified version of the scheme of Weiler and Flühler (2004).
The paper provides useful documentation of soil developmental processes over 13500 years. Together with an earlier study of Hartmann et al. (2020a), it provides evidence of the differences resulting from calcareous-vs.-siliceous parent materials.

The authors would like to thank John R. Nimmo for spending his time to review and make valuable comments to improve our manuscript. We highly appreciate the detailed feedback and recommendations to improve the consistency of our manuscript.
We will address the comments and suggestions below.

The data analysis is extremely thorough. A great variety of statistical methods are employed, perhaps more than necessary. I see little or no value in the Pdye analysis because the constraint of monotonicity is a serious shortcoming that could distort the interpretation of how water behaves in the profile.

We excluded the Pdye analysis in the revised manuscript, as the added value is only minor.

Presentation of multifactor comparisons of many individual experiments is unavoidably complex, and is done here (figures 3-10) through an organization that requires the reader's time and effort to understand and evaluate, but it does show the results in a way that the effects of soil age, irrigation intensity, and spatial variability can be directly seen.

The main problem I find in the manuscript is confusion and inconsistency concerning the classification of types of subsurface flow. Much of this relates to the term "finger flow", for which I don't find a clear definition in this paper, and which seems to be used in different ways.

Some background from my own understanding: Three main categories of preferential flow are commonly used—funneled flow, which is directed into particular downward paths as a result of heterogeneities of the medium that provide faster flowpaths through the more conductive material; fingered flow, which is initiated at flow instabilities in the wetting front and sustained in downward preferential paths by the greatly enhanced hydraulic conductivity of the newly wetted material; and macropore flow, which proceeds through elongated continuous pores over significant distances within the medium.

We understand the problem with a missing definition of the used terms. We included a more specific definition of the flow type classes in the revised manuscript on page 8, Line 3 to 35:

"[...] This classification method distinguishes between five flow types: (1) macropore flow with low interaction, (2) mixed macropore flow (low and high interaction), (3) macropore flow with high interaction, (4) heterogeneous matrix flow/finger flow, and (5) homogeneous matrix flow. The five flow types differ in the spatial extent of the water transport and thus in the proportion of the involved soil matrix, which impacts flow velocities, water availability and solute transport.  From (1) to (5) the preferentiality of the water transport decreases and the homogeneity and spatial water availability increases.
We define macropore flow as water transport via root channels, earthworm burrows, and flow along fissures largely bypassing the matrix. The characteristic dye patterns show narrow but long individual stains, which in some cases can be broader due to interactions with the surrounding soil matrix. The level of lateral interactions between the water transported via macropores and the surrounding soil matrix mainly depends on the soil matrix. In low permeable or saturated soils the lateral interactions are usually low and the characteristic dye patterns show narrow but long individual stains. Permeable soils enable high lateral interactions between macropore water and the soil matrix. The dye patterns show broader individual stains.
The term finger flow here summarizes all flow types that cause finger-shaped flow patterns, which includes finger flow caused by flow instabilities in the wetting front (Nimmo, 2021), finger-shaped flow paths due to water repellency, air entrapment or  textural layering (Hendrickx and Flury, 2001) and also funneled flow leading to vertical elongated finger-like flow paths. The latter is caused by the redirection and funneling of water by textural boundaries and large rocks (Hendrickx and Flury, 2001) or by the heterogeneity of soil hydraulic properties (Nimmo, 2021). The characteristic flow patterns of all these flow types are very similar and thus cannot be distinguished by the image analysis: they show broader, vertically elongated, coherent flow paths, which indicate a preferential vertical water transport and leave large parts of the soil matrix dry.
Dye patterns, which could not be classified as one of the five flow types were categorized as undefined.
We used a modified version (Hartmann et al., 2020a) of this classification which was more suitable for stony alpine soils. In the case of homogeneous matrix flow, the modified classification avoids that a high stone content leads to the detection of a heterogeneous flow pattern by breaking up the coherent stained area into smaller pieces, which then could be falsely classified as heterogeneous matrix flow, finger flow, or macropore flow depending on the abundance of rocks. In this case the flow type is assigned to a new flow type class called (6) 'homogeneous matrix flow between rocks'.
The modified classification also avoids a clear differentiation between 'macropore flow with high interaction' and 'finger-shaped flow'.  As the original classification assigns finger-shaped flow paths only when both, the medium-sized stained path width (20–200 mm) and the biggest stained path width class (> 200 mm) account for approximately half of the dye coverage, fingers with smaller widths (but not necessarily caused by macropores) were not detected as such and automatically counted as macropore flow with high interaction. Hartmann et al., 2020a observed that finger-like flow paths with smaller widths were frequently present in alpine soils. Their dye patterns and distributions of stained path width classes are similar to 'macropore flow with high interaction'.  Both classes cannot be distinguished from each other in the image analysis. Thus we renamed this class to (3) 'macropore flow with high interaction/ finger-shaped flow'. [...]"

I see these categories to be represented in the scheme of Weiler and Flühler (2004) (hereafter referred to as WF2004), which is designed specifically for use in interpreting dye-tracer results. Macropore flow needs matrix interaction to be visible, as acknowledged in the first three categories of WF2004. I see the term "matrix heterogeneous flow" as a synonym for funneled flow, and it is quite adequate in that

usage. Instability-initiated fingered flow would be difficult or impossible to distinguish from matrix heterogeneous flow when the only evidence is from pictures of dye-tracer distribution. Thus it is appropriate to group both of these flow modes together as in the fourth WF2004 category, "Heterogeneous matrix flow and fingering". Absence of preferential flow is reasonably called homogeneous matrix flow in the fifth category. In the present study, the use of the WF2004 classification scheme is a suitable approach for evaluating dye-tracer patterns in terms of preferential flow. It is extended reasonably with the added sixth category to accommodate effects of large stones in the soil.

The other modifications adopted here are poorly explained, and appear to deviate significantly from some widely understood general features of preferential flow, and from the evidence available from this study as I understand it. Below, I explain these issues further in relation to finger flow and macropore flow.

Finger flow
Instability-initiated fingers are possible, though my expectation in such heterogeneous soil is that these are likely to be rapidly channeled into funneled flowpaths. Based on the images and other available information in the present study, I doubt that it is possible to discern whether instability-initiated fingering is an active process. In 18:24 (location noted as page:line) the term "finger flow" seems to mean any preferential flow that is identified by finger-like patterns of dye tracer, not limited to the downward-moving fingers of wetness generated at a wetting-front instability. The finger-like patterns in the dye could result from other modes of preferential flow. If what is meant is just that the patterns have a finger-like shape, without regard to specific process, "finger flow" would be better replaced by the general term "preferential flow". This issue occurs also in 1:14, 21:4, 22:3-6, 22:16, 23:21-22, 24:5-8, and 26:4-13.

We agree that is was not made clear in the manuscript that we are talking about the shape of the flow pattern when referring to finger flow and not specifically to finger flow in the narrow sense (generated at a wetting-front instability). We addressed this issue by explaining our definition of the flow classes as written above and further included the following changes into the revised manuscript:

Page 1, Line 14: we changed "finger flow" to "finger-shaped flow paths"
The same changes were done on the following pages and lines: 1:11; 8:26-35; 19:12- 26; 22:4, 23:7-20; 24:9-26; 25:12, 20; 26:1-8; 27:18-24

On the other hand, the specific mode of instability-initiated finger flow is the subject of 23:4-6 and 24:11—25:2. It also is strongly related to the effects of hydrophobicity in 22:6 – 23:9. These passages need clarification and consistency. Overall, finger flow must be explicitly defined and the term used consistently. If the paper actually does claim that instability-generated finger flow is detected in these experiments, there needs to be justification for how this can be determined.

We see the point that using the term finger flow for finger-shaped flow paths without a previous definition of what we count as finger flow leads to misunderstandings and inconsistencies.
We replaced the term "finger flow" in the listed sections with the term "finger-shaped flow paths" to make it clear that we are talking not only specifically about finger flow in the narrow sense, but in the broad sense which includes all possible reason that could have caused the development of finger-shaped flow paths.

Macropore flow
There needs to be more discussion of the possible effects of macropores. The soils are likely rich in narrow macropores that result from growing and decaying roots (apparent in the images of both young and old soils), and other bioactive processes. If such macropores convey significant water that then has some degree of interaction with soil matrix material, they could create flow pattern features of the types observed. The statements in 18:15-22 are hard to understand and accept, where it is implied that finger flow can be distinguished from macropore flow, and stated that no macropore flow was found. If there are reasons to justify ruling out active macropore flow, they need to be carefully explained.

We see the point that we cannot rule out macropore flow in general, since macropores due to bioactivity (e.g. roots) are present and also very likely to conduct water. However, after carefully screening the photos, and from our observations during the excavation we cannot determine conclusively whether the staining pattern is a result of macropore flow with high interactions or whether finger-shaped flow paths, caused by a variety of site conditions, obscure the water transport in existing macropores. Thus, we changed the statement about the possibility of an impact of macropore flow in the revised manuscript on page 19, line 13-21

"[..] At the youngest moraine matrix flow is the predominant flow type (relative frequency > 0.6) followed by the flow type class 'Macropore flow with high interaction/ Finger-shaped flow paths'. A reliable distinction between macropore flow with high interaction and finger-shaped flow could neither be made through the image analysis nor through on-site assessment. As narrow macropores were sometimes present (e.g. thin root channels), they certainly also contribute to water transport, but it is also likely that this process is obscured by finger-shaped flow paths. Since the water transport patterns of both flow types cannot be distinguished and show finger-shaped flow patterns, they are also referred to as finger-shaped flow in the following. [..]"

I cannot make sense of the statements in 7:30-33, which seem to imply that finger flow can be distinguished from macropore flow, but then contradict that in saying that no such differentiation is made. Then there is confusion in the statement that narrow finger flowpaths could somehow be misclassified as macropore flow with high (but not low or intermediate) interaction.

We clarified the rationale for this joint flow type class, as described above. We also corrected the statement that macropore flow could be ruled out (see above) and we included the statement that both flow class types cannot be distinguished from each other in the image analysis.

We clarified on page 8, line 28-35:
"[..] The modified classification also avoids a clear differentiation between 'macropore flow with high interaction' and 'finger flow'.  As the original classification assigns finger-shaped flow paths only when both the medium-sized stained path width (20–200 mm) and the biggest stained path width class (> 200 mm) account for approximately half of the dye coverage, fingers with smaller widths (but not necessarily caused by macropores) were not detected as such and automatically counted as macropore flow with high interaction. Hartmann et al., 2020a observed that finger-like flow paths with smaller widths were frequently present in alpine soils. Their dye patterns and distributions of stained path width classes are similar to 'macropore flow with high interaction'. Both classes cannot be distinguished from each other in the image analysis. Thus we renamed this class to (3) 'macropore flow with high interaction/ finger flow' [...]".

Section 3.2 (18:12-30) needs to be rewritten for consistency with other clarifications. The category "Macropore flow with high interaction/ Finger flow" is mentioned here and in Figure 10, but it is not mentioned in the definition of the categories on page 7 and is not in the scheme of WF2004.

The class "Macropore flow with high interaction/ Finger flow" is part of an adapted version of the scheme of WF2004 by Hartmann et al., 2020 to also include smaller sized finger-shaped flow paths. We included this clarification into the revised manuscript as stated above.
In the revised manuscript we rewrote section 3.2 on page 19, line 11-32

"[…] Using the VD-profiles of the three SPW classes and their fraction of the total dye coverage to characterize flow types (Weiler, 2001) we found that over the millennia flow types transition from matrix flow to preferential flow in form of finger shape flow paths (Figure 10 a).
At the youngest moraine matrix flow is the predominant flow type (relative frequency > 0.6) followed by the flow type class 'Macropore flow with high interaction/ Finger flow'.
A reliable distinction between macrpore flow with high interaction and finger-shaped flow could neither be made through the image analysis nor through on-site assessment. As narrow macropores were sometimes present (e.g. thin root channels), they certainly also contribute to water transport, but it is also likely that this process is obscured by finger-shaped flow paths. Since the water transport patterns of both flow types cannot be distinguished and both show finger-shaped flow patterns, they are also referred to as finger-shaped flow in the following.
At the 160a moraine the relative frequency of matrix flow decreased to 0.5 and the frequency of finger-shaped flow increased. At the two oldest moraines the dominant flow type is flow with finger-shaped flow paths and the relative frequency of matrix flow dropped below 0.3.
Considering the entire profile depth of 1 m, the frequency of matrix flow decreases and the frequency of finger shape flow paths increase continuously with moraine age. A depth differentiated view shows a higher proportion of finger-shaped flow at the 4.9ka than at the 13.5ka in the upper 20 cm (Figure 10 a1). In the other depths (Figure 10 a2 to a4), however, a continuous increase in finger-shaped flow frequency with moraine age was observed. With regard to the irrigation intensity no consistent impact on the flow type distribution across the millennia could be identified (Figure 10 b). At the 110a and 160a moraine the two dominant flow types (matrix flow and finger-shaped flow paths) show an almost equal distribution across all irrigation intensities. A tendency to less matrix flow is observed at the 4.9ka, whereas at the 13.5ka the frequency of matrix flow increases with increasing irrigation intensity. Differentiated by depth, we observed no systematic trend in flow type frequency distribution with increasing irrigation intensity in the upper 20 cm for all age groups (Figure 10 b1). Below a depth of 20 cm, the 4.9ka and 13.5ka each show a trend-like behavior in the shift of the frequency distribution with irrigation intensity similar to what we see for the entire soil profile (Figure 10 b2 to b4). Below a depth of 40 cm, the relative frequency of matrix flow also increases with increasing irrigation intensity at the 110a and 160a (Figure 10 b3 to b4). […]"

Overall:
This paper is dense with useful information and provides insights into the development of preferential flow paths during landscape evolution and several other important facets of unsaturated flow in calcareous soils. It needs revision for consistency and adherence to evidence and general understanding of the different types of preferential flow paths. Because the basic experimental work and presentation of data are sound, I have classed these revisions as minor, though I see them as extremely important.

We are happy to hear this positive assessment of our work and have made the suggested improvements.

Specific Comments

 Rewrite for clarity. Use of "below" in line 16 suggests that the excavation is downward to produce horizontal planes, but "vertical profiles" in 17 suggests otherwise. Does "below" mean "downslope of"? The operation suggests that a trench was first excavated off to the side of the plot to provide access for vertical profiling. More details on this would be helpful.

We clarified in the revised manuscript on page 7, line 8:
"A first vertical profile was excavated 10-15 cm downslope of the lower edge of the irrigated plot to check for subsurface lateral flow."

7:17. What is meant by "amount"? The number of flow paths?

We clarified in the revised manuscript on page 7, line 30:
"The surface area density (SAD) is an indicator for the number of individual flow paths and was calculated for each pixel row of the five profiles by using the intercept density, which describes the number of interfaces between stained and unstained pixels divided by the horizontal width of the soil profile."

7:28-29. Clarify—maybe make two sentences. Start with a clear description of the problem caused by rocks. Then the solution devised.

We clarified in the revised manuscript on page 8 line 22-28:
"We used a modified version (Hartmann et al., 2020a) of this classification which was more suitable for stony alpine soils. In case of homogeneous matrix flow, the modified classification avoids that a high stone content leads to the detection of a heterogeneous flow pattern by breaking up the coherent stained area into smaller pieces, which then could be falsely classified as heterogeneous matrix flow, finger flow, or macropore flow depending on the abundance of rocks. In this case the flow type is assigned to a new flow type class called (6) 'homogeneous matrix flow between rocks'."

7:31-33 Why "misclassified"? What is unreasonable about "macropore flow with high interaction"?

As stated above, we removed this statement and clarified in the revised manuscript on page 8, line 28-35 as follows:

"The modified classification also avoids a clear differentiation between 'macropore flow with high interaction' and 'finger flow'. As the original classification assigns finger-shaped flow paths only when both the medium-sized stained path width (20–200 mm) and the biggest stained path width class (> 200 mm) account for approximately half of the dye coverage, fingers with smaller widths (but not necessarily caused by macropores) were not detected as such and automatically counted as macropore flow with high interaction. Hartmann et al., 2020a observed that finger flow paths with smaller widths were frequently present in alpine soils. Their dye patterns and distributions of stained path width classes are similar to 'macropore flow with high interaction'. Both classes cannot be distinguished from each other in the image analysis. Thus we renamed this class to (3) 'macropore flow with high interaction/ finger flow'."

7:34. Proportion in relation to what? PFF needs to be defined more clearly.

We clarified in the revised manuscript on page 9, line 1-4:

"To quantify the proportion of preferential flow per profile, a preferential flow fraction index (PFF) was calculated as the proportion of all preferential flow type classes at each profile. As the flow type classification was done for each pixel row, the PFF is the number of pixel rows classified as a preferential flow type (flow types (1)-(4)) divided by the total number of pixel rows."

We clarified in the revised manuscript on page 13, line 6-7:

"At the 4.9ka and 13.5ka the fraction of SPW>200 mm is lower in the upper 10-20 mm compared to the young moraines.  Over the entire profile range, path widths of the category 20<SPW<200 mm most often have the largest share of the dye coverage."

14:1. Replace "Whereas" with "In contrast," or similar expression.

We updated the sentence in the revised manuscript on page 15, line 1 to:

"In contrast, the old moraines have a high fraction of 20<SPW<200mm combined with a high SAD, which indicates a higher number of smaller, narrow blue-colored areas and thus more individual active flow paths at the older moraines and less individual flow paths but larger continuous areas used for water transport at the young moraines."

17:4-5. It seems at best to be a very subtle effect for the middle portion of the profiles to be less significantly different. Maybe not worth mentioning.

We excluded this statement in the revised manuscript.

We updated the sentence in the revised manuscript on page 22, line 35 to:

"Only larger stones and occasional clay lenses (the size of a few centimeters) or other material heterogeneities create heterogeneous matrix flow."

We updated the sentence in the revised manuscript on page 23, line 6 to:

"At both moraines deep infiltration, almost no surface runoff, and no subsurface lateral flow was observed."

22:4-5. The point is not about the water transport in general but the pattern of the water transport that is affected. Insert "pattern" or some similar expression.

We updated the sentence in the revised manuscript on page 23, line 8 to:

"Thus, water infiltrated heterogeneously and/or the water transport pattern was affected by properties of the soil surface or of the upper soil layer."

We updated the sentence in the revised manuscript on page 24, line 25-27 to:

"After more than 10000 years of landscape development, subsurface hydrology at the calcareous geology is ruled by finger-shaped flow and deep infiltration, whereas at the siliceous geology storage capacity in the top soil strongly increased with a corresponding reduction in infiltration depths and a shift to macropore flow."

We corrected "Matrix potential" to "Matric potential" in the revised manuscript on page on page 24, line 34.

23:32. The paragraph starting here, and also the next one, are all about the two older soils. This should be made clear to the reader in the first sentence at line 32. Consider rearranging discussion from this point through 25:13 in order to proceed in the logical order of young to old.

We rearranged the discussion from page 23, line 32 to page 25, line 13 in the logical order of young to old:

"At the young moraines (110a and 160a), we observed an in increase in the frequency of matrix flow at greater depths with increasing irrigation intensity, which is caused by an increase in the SPW>200 mm with a simultaneous decrease in the median dye coverage and a decrease in the number of flow paths (SAD) (Figure 5e). The decrease in median dye coverage with increasing intensity is particularly pronounced at the four bare plots of these young moraines (data not shown). No clear trend can be seen at the two plots with a higher vegetation cover. However, the decrease in SAD and the increase in stained path widths indicates that water flow paths that reach greater depths tend to widen and to merge with increasing irrigation intensity. This process might be facilitated by higher water contents at greater depths or by a change in material properties.
At the 4.9ka moraine, we also observed an increase in dye coverage (Figure 5 (a)) and an increase in the number of flow paths (Figure 5 (e)). The proportion of 20<SPW<200 mm increases (Figure 5 (c)) and the proportion of SPW>200 mm decreases (Figure 5 (d)), which then leads to an increase in the frequency of finger flow paths in the flow type classification (Figure 10 (b2) to (b4)). At the 13.5ka we observed an increase in dye coverage (Figure 5 (a)), an increase in infiltration depth (Figure 3), a broadening of the flow paths (Figure 5 (c-d)), and an increase in the number of flow paths (Figure 5 (e) and Figure 6) with

increasing irrigation intensity. Different to the 4.9ka, the increase in the proportion of SPW>200 mm on the dye coverage leads to a transition to more matrix flow (Figure 10) in the flow type classification. The impact of irrigation intensity on water flow paths is slightly obscured by the process of flow type classification, as it is only based on the occurrence of the three SPW classes as fractions of the dye coverage (Weiler, 2001). The number of flow paths or the dye coverage itself are not taken into account. We observed at both age classes that with increasing irrigation intensity more fingers are generated and more soil space is used for water transport (Figure 5). The formation of finger flow paths and their properties such as number, flow velocity, or width are in a complex interplay with the surrounding soil moisture, the flux rate and soil properties (Nimmo, 2021).

Studies focusing on the formation of finger-shaped flow paths found the finger width not only to be influenced by soil properties, and initial and boundary conditions (Glass et al., 1989), but also by the flow rate through the finger (Parlange and Hill (1976), White et al. (1976)) with higher flow rates leading to an increase in finger width. This was also observed by Ma et al. (2008), who also found a positive correlation between rainfall intensity, time of finger flow occurrence and mean velocity. The increase in mean velocity of the fingers leads to a faster downward transport and thus deeper infiltration depths with higher irrigation intensities (Cremer et al., 2017). An increase in the number of fingers with higher flux rates was also observed (Sililo and Tellam, 2000). These findings by other studies are similar to our observations at the 13.5ka moraine. It is unclear what causes the different observations in the dominant flow path widths at the 4.9ka and 13.5ka moraines. We can only speculate whether the higher organic matter content, the higher root density, or soil properties such as the lower hydraulic conductivities and higher porosity play a role in producing narrower flow paths with increasing irrigation intensity at the 4.9ka moraine."

**25:24-27. Confusing. Which of the plots were less affected by the direct application of water? Why is there consideration of the boundaries in this?**

We mention the observations of the deep infiltration at the plot boundaries of the bare plots at both young moraines to support the assumption that water would have infiltrated deep into the soil, when the soil surface was not affected by structural sealing due to the irrigation process.

We rephrased this part in the revised manuscript on page 27, line 5-14 for clarification:

"We further assume that the irrigation with the hand-operated sprayer, which had to be held close to the soil surface due to strong winds, sometimes led to a high force of application and promoted structural sealing at the bare plots of the 110a and 160a moraines. At both moraines, deep infiltration was often found at the boundaries of the bare plots. Since the plot boundaries were not irrigated, they were also not affected by structural sealing. Water running off to the sides infiltrated deep into the soil. This observation suggests that a more homogeneous and deep transport of the water can take place in this quite homogeneous and unsorted material (Hartmann et al., 2020b), if the surface is not influenced by particle displacement. Thus, it is assumed that the proportion of preferential flow paths at the young moraines is generally overestimated and homogeneous to heterogeneous matrix flow with deep infiltration are probably the dominant flow types under natural rainfall conditions. As the plot boundaries (outer boundaries and boundaries of neighboring subplots) are excluded from the image analysis to avoid edge effects, the here observed deep percolation could not be accounted for in our quantitative analysis."

---

## Referee Report (RR1)

Revised manuscript—review by John Nimmo.

This manuscript is well-revised should be published in HESS. Especially important is the added and clarified material on flow types in sections 2.3 and 3.2. As to its value, I agree with my previous statement: "This paper is dense with useful information and provides insights into the development of preferential flow paths during landscape evolution and several other important facets of unsaturated flow in calcareous soils."

Before publication, I recommend only minor changes, listed below.

8:11  Should be "low-permeability or ".

8:12  Replace "low" with "minimal" and "high" with "extensive".

8:20  Replace "transport and leave" with "transport, leaving".

8:22  Replace "this" with "the Weiler and Flühler".

8:27  Delete "both".

8:33  Replace "Both" with "These".

13:10  "at the 110a moraine"

13:17  Plural: "110a and 160a moraines."

14:6  Replace "In" with "At".

15:13  Unclear sentence: "In case of SAD trends differed between the age classes."

19:15  Word order: "could be made neither through"

19:23  Change word: "At the other depths"

19:23-32  The expression "at the XXa" should be either "at XXa" or "at the XXa moraine"

22:12  Maybe "dense", or "high-density" instead of "high".

23:12  Authors here should not be in parentheses.

25:4  Not "caused by" but "evidenced by."

25:15  Should be "Different from".

25:15 (and elsewhere)  Flux is a rate, so it should be just "the flux" not "the flux rate".

27:14  "quantitative"

27:32  "major" would be better than "primary". Or else delete "only".

27:31-28:5  The new material here should be rewritten to flow more smoothly. There are too many uses of "therefore". Many of the factors here can be stated without implying causal relationships among them. And the transfer functions seem to come out of nowhere—maybe omit mention, or connect them more to your approach or findings.